# Smc5/6's multifaceted DNA binding capacities stabilize branched DNA structures

Jeremy T-H. Chang [1,2,5], Shibai Li[3,5], Emily C. Beckwitt[4], Thane Than[3], Cory Haluska [3], Joshua Chandanani[1], Michael E. O'Donnell [4], Xiaolan Zhao [2,3] ✉ & Shixin Liu [1,2] ✉

Smc5/6 is an evolutionarily conserved SMC complex with roles in DNA replication and repair, as well as in viral DNA restriction. Understanding its multiple functions has been hampered by a lack of mechanistic studies on how the Smc5/6 complex associates with different types of DNA. Here we address this question by simultaneously visualizing the behavior of Smc5/6 on three types of DNA, namely double-stranded (ds) DNA, single-stranded (ss) DNA, and junction DNA formed by juxtaposed ss- and dsDNA, using correlative single-molecule fluorescence and force microscopy. We find that Smc5/6 displays distinct behaviors toward different types of DNA, dynamically associating with dsDNA while stably binding to junction DNA. Mechanistically, both the Nse1-3-4 subcomplex and ATP binding enhance the complex's dsDNA association. In contrast, Smc5/6's assembly onto ssDNA emanating from junction DNA, which occurs even in the presence high-affinity ssDNA binders, is aided by Nse1-3-4, but not by ATP. Moreover, we show that Smc5/6 protects junction DNA stability by preventing ssDNA annealing. The multifaceted DNA association behaviors of Smc5/6 provide a framework for understanding its diverse functions in genome maintenance and viral DNA restriction.

Structural maintenance of chromosomes (SMC) complexes are primordial DNA-interacting complexes critical for the organization and maintenance of the genome. In eukaryotes, the cohesin and condensin SMCs fold and tether DNA, while the Smc5/6 complex (Smc5/6 in short) directly facilitates DNA replication and repair and restricts episomal viral DNA[1]. Conserved roles of Smc5/6 include the regulation of stalled or damaged replication forks and the control of recombinational repair structures[2]. A common feature of these DNA replication and repair intermediates is a junction DNA architecture where segments of ssDNA and dsDNA are juxtaposed to each other. Addressing how Smc5/6 interacts with and modulates junction DNA in comparison with its association with dsDNA and ssDNA can provide a mechanistic understanding of its multiple functions.

Biochemical data suggest that Smc5/6 binds to ssDNA and dsDNA via different constituents. Among its eight subunits, the heterodimer of Smc5 and Smc6 forms the tripartite backbone of the complex onto which six non-Smc subunits (NSEs) attach (Fig. 1a)[3–6]. In the budding yeast Smc5/6, the Nse1, Nse3, and Nse4 subunits form a subcomplex (Nse1-3-4) that binds to dsDNA, while Nse2, Nse5 and Nse6 do not exhibit direct DNA binding activity[3,7]. Additional DNA binding capacity is offered by Smc5 and Smc6. These two SMC subunits contain globular hinge and head regions connected via long coiled-coil arm regions (Fig. 1a)[3–6]. Their dimerized hinge regions exhibit ssDNA binding ability, whereas their head regions bind dsDNA and dimerize upon binding to ATP[7,8]. The thus-formed head dimer (engaged head, or E-head) was recently shown to associate with Nse1-3-4, forming a DNA clamp

[1]Laboratory of Nanoscale Biophysics and Biochemistry, The Rockefeller University, New York, NY 10065, USA. [2]Tri-Institutional MD-PhD Program, The Rockefeller University, Weill Cornell Medical College, and Memorial Sloan Kettering Cancer Center, New York, NY 10065, USA. [3]Molecular Biology Program, Memorial Sloan Kettering Cancer Center, New York, NY 10065, USA. [4]Laboratory of DNA Replication, Howard Hughes Medical Institute, The Rockefeller University, New York, NY 10065, USA. [5]These authors contributed equally: Jeremy T-H. Chang, Shibai Li. ✉e-mail: zhaox1@mskcc.org; shixinliu@rockefeller.edu

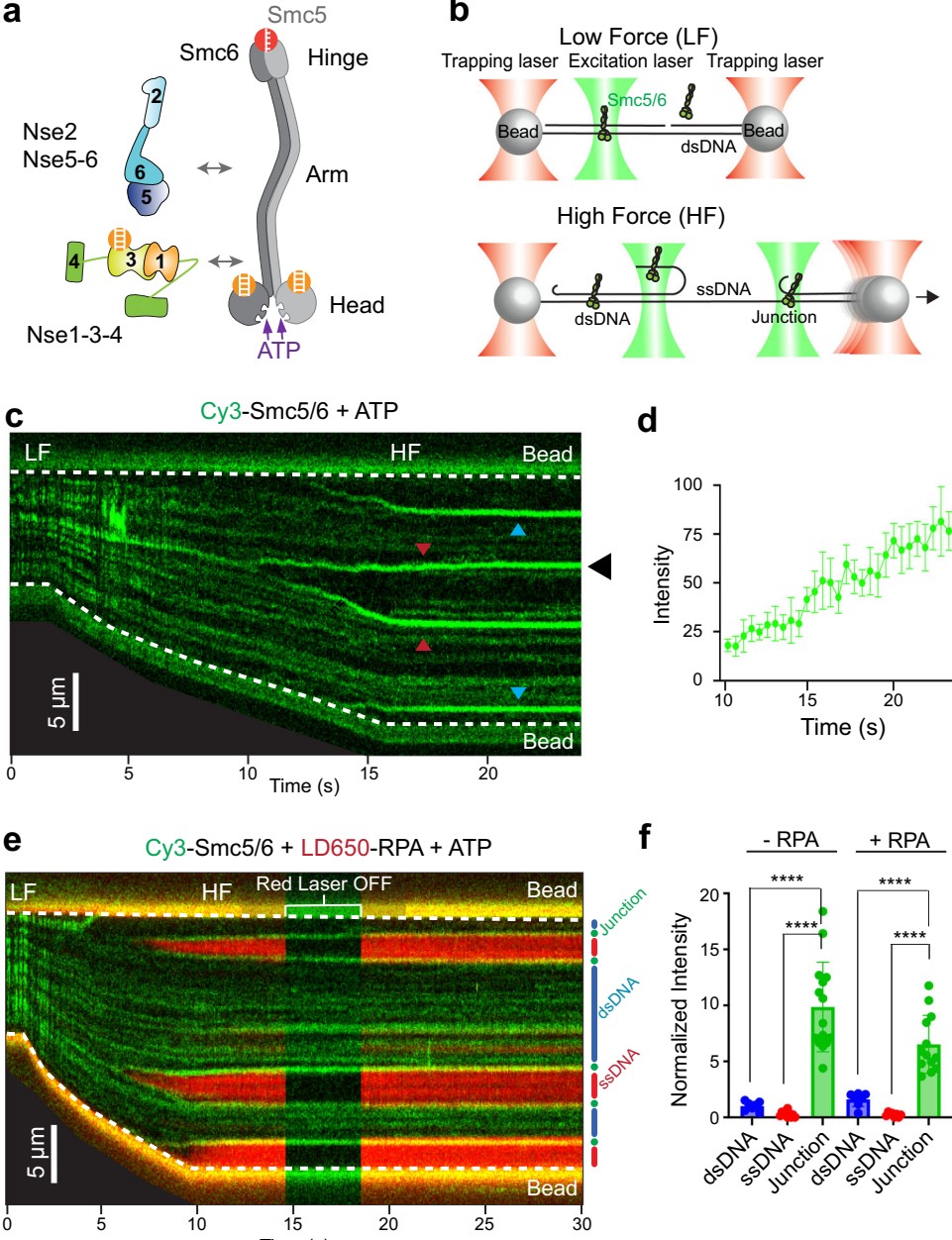

**Fig. 1 | Smc5/6 exhibits distinct binding behaviors on three types of DNA.**
**a** Architecture of the *S. cerevisiae* Smc5/6 complex, which is composed of a Smc5-Smc6 heterodimer and six other subunits (Nse1-6). Orange symbols indicate regions of Smc5/6 known to bind dsDNA. The red symbol indicates a region known to bind ssDNA. **b** Schematic of the single-molecule experimental setup under a low-force (LF) or high-force (HF) regime. Multiple forms of DNA co-existed on the same tether in the HF regime due to force-induced DNA peeling. The DNA binding behavior of fluorescently labelled proteins was monitored by confocal microscopy. **c** Representative kymograph of a λ DNA tether being stretched from LF to HF in the presence of 20 nM Cy3-Smc5/6 (green) and 2 mM ATP. Red arrows indicate twin-streaks emerging from an internal DNA nick, and blue arrows indicate single streaks emerging from untethered DNA termini. **d** Quantification of the Smc5/6 fluorescence signals at a junction DNA site (marked by the black arrow in panel **c**) over time. Data points indicate the averaged photon count per frame (*n* = 10 frames) at

the junction DNA site and error bars represent standard deviation. **e** Representative kymograph of a DNA tether being stretched from LF to HF in 20 nM Cy3-Smc5/6 (green), 10 nM LD650-RPA (red), and 2 mM ATP. Smc5/6 and RPA were illuminated by two excitation lasers (532 nm and 638 nm, respectively). The red laser was turned off momentarily to confirm signals from Smc5/6. dsDNA (blue), ssDNA (red), and junction DNA (green) regions of the tether are indicated to the right of the kymograph. **f** Smc5/6 fluorescence signals on different types of DNA under HF. Bar heights indicate the group mean and error bars represent standard deviation. *P* values were determined from two-tailed unpaired *t*-tests with Welch's correction (****$P < 0.0001$). Sample sizes are: dsDNA −RPA (*n* = 7); ssDNA −RPA (*n* = 10); Junction −RPA (*n* = 15); dsDNA + RPA (*n* = 7); ssDNA + RPA (*n* = 8); and Junction + RPA (*n* = 13), where *n* indicates the number of regions analyzed. Source data for panels **d** and **f** are provided within the Source Data file.

---

capable of encircling a DNA duplex (Supplementary Fig. 1a)[7]. In this structure, Nse3 and the Smc5-6 head regions embrace dsDNA from opposite sides. In another structure obtained without ATP or DNA, Nse3 is positioned abutting the Smc5 head region (Supplementary

Fig. 1b)[9]. This configuration brings two dsDNA binding modules adjacent to each other. These snapshots of Smc5/6 structures suggest that the complex can adopt at least two conformations with the main dsDNA binding constituents positioned either on opposite sides

(with ATP) or next to each other (without ATP). How ATP quantitatively influences the dynamics of Smc5/6 on different forms of DNA has been unclear.

Building upon biochemical and structural data, we investigated how Smc5/6 interacted with different types of DNA, and the effects of ATP and Nse1-3-4 on the complex's DNA binding abilities. We sought to understand how Smc5/6 engages with junction DNA that resembles stalled or damaged replication forks, since Smc5/6 has been shown to be enriched at these sites in cells. We also aimed to understand how the association of Smc5/6 with junction DNA could affect the behaviors of both the complex and the DNA. To address these questions, we employed single-molecule techniques that could directly visualize protein complexes on different types of DNA[10]. Time-resolved single-molecule fluorescence microscopy in combination with optical tweezers allowed us to simultaneously evaluate the dynamics of fluorescently labelled Smc5/6 on ssDNA, dsDNA, and junction DNA. Our data revealed an unexpected dichotomous behavior of Smc5/6 in association with linear dsDNA versus junction DNA as well as Smc5/6 assembly onto free ssDNA. We uncovered a contribution of ATP in the topological entrapment of dsDNA by Smc5/6 as well as contributions of Nse1-3-4 toward DNA association. Finally, we observed that Smc5/6 stabilizes junction DNA, which could be explained by its multifaceted DNA binding behavior. Collectively, the visualization and dissection of Smc5/6's DNA association properties at the single-molecule level provide a foundation to comprehend how Smc5/6 can move along linear dsDNA while accumulating at and regulating branched DNA structures such as replication forks and DNA repair intermediates.

## Results
### Experimental design
We employed dual-trap optical tweezers combined with scanning confocal microscopy to examine Smc5/6 interactions with DNA (Supplementary Fig. 1c). We used the bacteriophage λ genomic DNA (48.5 kbp in length) as a model DNA substrate. Each of the two ends of the DNA was biotinylated on one strand and conjugated to a streptavidin-coated bead trapped by a focused infrared laser (Fig. 1b)[11]. The force applied to the λ DNA tether was controlled by changing the distance between the two beads, with a larger separation corresponding to a higher force. We conducted our experiments at two force regimes: a low-force (LF) regime at ~5 picoNewtons (pN) wherein the DNA tether was held taut to facilitate single-molecule imaging and remained in the B-form duplex structure, and a high-force (HF) regime at ~60 pN wherein parts of the duplex DNA melted into separated ssDNA from a few (normally one to three) randomly positioned internal nicks or from the untethered DNA ends (Fig. 1b)[12,13]. The in situ generated ssDNA was adjacent to a dsDNA segment, thereby forming a ss-dsDNA junction (Fig. 1b). This experimental design permitted simultaneous evaluation of Smc5/6's binding behaviors on multiple types of DNA: linear dsDNA and ssDNA, and ss-dsDNA junctions.

The eight subunits of the budding yeast Smc5/6 complex were co-overexpressed, and the complex was purified as described (Supplementary Fig. 2a)[3]. The tags added to Smc5 and Nse6 for the purpose of purification did not interfere with Smc5/6 functions, since the fusion constructs supported cell growth when they were used to replace the untagged counterparts (Supplementary Fig. 2b). Visualization of Smc5/6 was achieved by either site-specific enzymatic labeling by Cy3 or non-specific labelling using AlexaFluor555 as described before[14]. In the former, a S6-peptide tag was appended to the C-terminus of Smc5 for single-fluorophore labeling. In the latter, multiple fluorophores were placed on the N-termini of any subunits without additional tagging to ease the comparison among different forms of the complex. The two labeling strategies produced consistent results as detailed below. As seen previously, the purified wild-type Smc5/6 complexes exhibited ATPase activities and an ATP-dependent ability to trap circular dsDNA (Supplementary Fig. 2c, d)[4].

## Smc5/6 exhibits distinct binding behaviors on dsDNA, ssDNA, and ss-dsDNA junctions
To visualize the behavior of Smc5/6 on different types of DNA, the λ DNA tether was moved to a channel containing Cy3-labelled Smc5/6 complexes in the presence of saturating ATP (2 mM). The position and intensity of Smc5/6 signals along the entire DNA tether were monitored while the DNA was being stretched. When the DNA tether was held at a low tension (dsDNA$_{LF}$), we observed Smc5/6 binding across the tether without noticeable positional dependence (Fig. 1c, Supplementary Movie 1). Strikingly, when the DNA tether was stretched to generate ssDNA, and consequently ss-dsDNA junctions, streaks of Smc5/6 with stronger fluorescent signals than those seen on dsDNA emerged, either as single streaks proximal to the beads (Fig. 1c, blue arrows) or as twin-streaks initiated from internal DNA nicks (Fig. 1c, red arrows). The fluorescence intensities at these streaks increased as the inter-bead distance became larger (Fig. 1d). Based on previous studies, such streak signals reflect protein accumulation at the ss-dsDNA junctions and/or on the adjacent free ssDNA created by force-induced peeling[12].

Low levels of dynamic Smc5/6 signals were observed inside the regions between twin-streaks or between the streak close to the DNA end and the proximal bead, both of which correspond to stretched ssDNA regions (Fig. 1c). Diffusive Smc5/6 binding was also detectable on dsDNA regions outside twin-streaks under high force (Fig. 1c). Quantification of the fluorescence signals showed that Smc5/6 was significantly enriched at DNA junction sites compared to dsDNA and ssDNA regions (Fig. 1f).

To validate the classification of different DNA regions, we introduced the eukaryotic RPA complex that binds to ssDNA but not dsDNA[15]. Two-color imaging using Cy3-labelled Smc5/6 and LD650-labelled RPA showed that the RPA-coated ssDNA regions were indeed flanked by Smc5/6 streak signals (Fig. 1e), confirming that these streaks were at DNA junction sites. The preferential binding of Smc5/6 to DNA junctions over dsDNA regions was still observed in the presence of RPA, although the Smc5/6 intensity at the junctions was somewhat lowered perhaps due to the competition between Smc5/6 and RPA for ssDNA binding (Fig. 1f).

## ATP enhances Smc5/6 association with dsDNA, but not with DNA junctions
Like other SMC complexes, Smc5/6 is an ATP binding and hydrolyzing molecular machine. Therefore, we sought to understand the role of ATP in Smc5/6 association with different types of DNA substrates. To this end, the same set of experiments described above were conducted in the absence of ATP. The most striking difference observed was a significant reduction of Smc5/6 signals at dsDNA regions at both low and high force conditions (Fig. 2a, b). Despite reduced signals, Smc5/6 binding to dsDNA was still observed in the absence of ATP, especially at the low force regime (Fig. 2a, b). This observation indicated that Smc5/6 has two dsDNA binding modes—one ATP-independent and the other ATP-dependent. ATP enhancement of Smc5/6 binding to dsDNA seen here at the single-molecule level is consistent with biochemical data showing that ATP promotes Smc5/6's topological entrapment of circular dsDNA (Supplementary Fig. 2c)[4,16] and a recent cryo-EM structure showing that ATP-bound Smc5/6 can form a clamp on duplex DNA[7].

In contrast to the ATP dependence of Smc5/6 binding to dsDNA, Smc5/6 association with DNA junctions was unaffected by ATP as similar levels of Smc5/6 signals were seen regardless of ATP (Fig. 2c). This was confirmed by two-color imaging experiments using LD650-RPA to distinguish ssDNA, dsDNA, and junction DNA along the DNA tethers (Fig. 2d, e). The differential effects of ATP on Smc5/6 binding to linear dsDNA versus junction DNA (which additionally contains a ssDNA flap) suggested that Smc5/6 associates with these substrates via distinct mechanisms. As the only known ssDNA binding region in

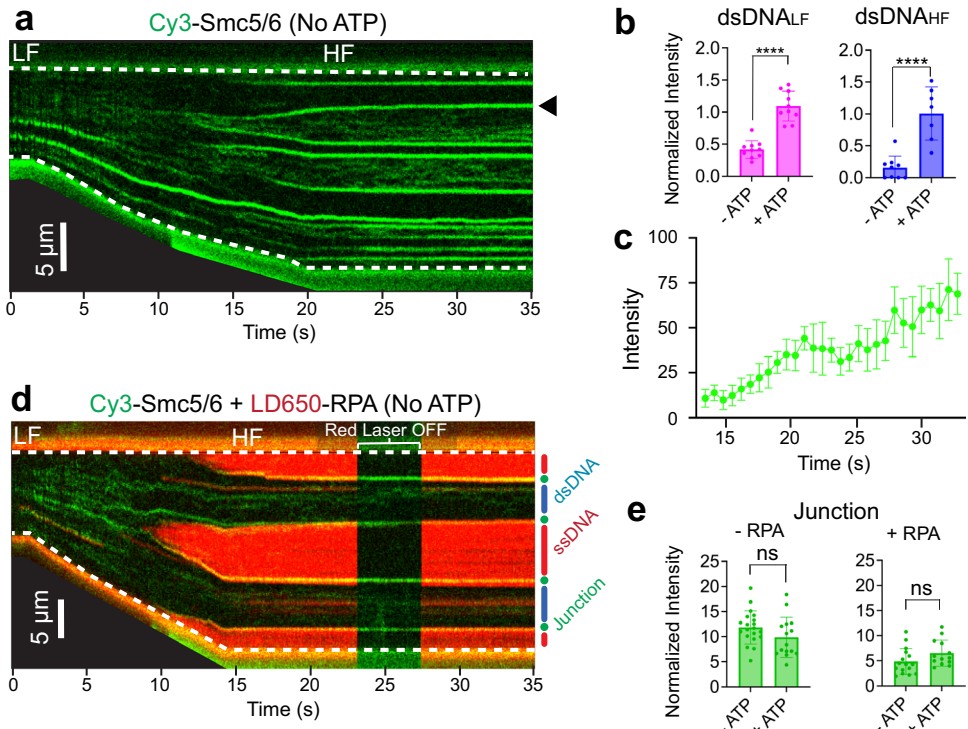

**Fig. 2 | Smc5/6 exhibits ATP-dependent binding to dsDNA and ATP-independent binding to junction DNA. a** Representative kymograph of a λ DNA tether being stretched from LF to HF in the presence of 20 nM Cy3-Smc5/6 (green) without ATP. **b** Comparison of Cy3-Smc5/6 fluorescence signals on dsDNA between −ATP and +ATP conditions under LF or HF (normalized by the number of pixels for each region). Bar heights indicate the group mean and error bars represent standard deviation. *P* values were determined from two-tailed unpaired *t*-tests (****$P < 0.0001$). Sample sizes are: dsDNA$_{LF}$ − ATP ($n = 10$); dsDNA$_{LF}$ + ATP ($n = 10$); dsDNA$_{HF}$ −ATP ($n = 9$); and dsDNA$_{HF}$ + ATP ($n = 7$), where *n* indicates the number of regions analyzed. **c** Quantification of the Smc5/6 fluorescence signals at a junction DNA site (marked by the black arrow in panel **a**) over time in the absence of ATP. Data points indicate the averaged photon count per frame ($n = 10$ frames) at the junction DNA site and error bars represent standard deviation. **d** Representative

kymograph of a λ DNA tether being stretched from LF to HF in the presence of 20 nM Cy3-Smc5/6 (green) and 10 nM LD650-RPA (red) without ATP. dsDNA (blue), ssDNA (red), and junction DNA (green) regions of the tether are indicated to the right of the kymograph. **e** Comparison of the Cy3-Smc5/6 fluorescence signals at junction DNA sites between −ATP and +ATP conditions in the absence or presence of RPA. Bar heights indicate the group mean and error bars represent standard deviation. *P* values were determined from two-tailed unpaired *t*-tests (ns, not significant). $P = 0.1173$ for −RPA condition, and $P = 0.0956$ for +RPA condition. Samples sizes are: Junction −RPA −ATP ($n = 20$); Junction −RPA + ATP ($n = 15$); Junction + RPA −ATP ($n = 16$); Junction + RPA + ATP ($n = 13$), where *n* indicates the number of regions analyzed. Source data for panels **b**, **c** and **e** are provided within the Source Data file.

Smc5/6 is the hinge domain[8], we infer that the hinge likely interacts with ssDNA regardless of ATP.

The distinct behaviors of Smc5/6 on dsDNA versus at junction DNA were also reflected in the different kinetics of its dissociation from these substrates (Supplementary Fig. 3). In the presence of ATP, a population of Smc5/6 complexes remained bound to dsDNA and underwent sliding along the DNA after the tether was moved to a buffer-only channel (Supplementary Fig. 3a). In the absence of ATP, almost all of the Smc5/6 complexes dissociated immediately from dsDNA, and the few sliding complexes lasted for much shorter times on the DNA compared to the +ATP condition (Supplementary Fig. 3b, c). Notably, the sliding Smc5/6 complexes observed in the presence of ATP could persist under a high-salt (500 mM NaCl) wash (Supplementary Fig. 4), providing evidence for a topologically entrapped species as suggested by previous studies[4,7,16]. In contrast, Smc5/6 signals at junction DNA generated under high force lasted significantly longer than on dsDNA regardless of whether ATP was present (Supplementary Fig. 3d, e), supporting the notion that Smc5/6 stably and preferentially associates with junction DNA.

## ATP hydrolysis is not required for enhanced Smc5/6 association with dsDNA
Next, we examined whether ATP hydrolysis by Smc5/6 is required for the ATP-mediated enhancement of dsDNA binding. To this end, we

purified an Smc5/6 complex containing Walker B motif mutations (Smc5-E1015Q, Smc6-E1048Q) that impaires ATP hydrolysis but not ATP binding (Supplementary Fig. 2a). We confirmed previous findings that this mutant complex (referred to as the Smc5/6 ATPase mutant) has a diminished ATPase activity but retains the DNA entrapment ability (Supplementary Fig. 2c, e)[4,16]. The Smc5/6 ATPase mutant was nonspecifically labelled with AlexaFluor555. The wild-type complex labelled in this fashion (referred to as Smc5/6 8-mer) displayed the same behavior in terms of differential binding to ss-, ds-, and junction DNA as the site-specifically labelled Cy3-Smc5/6 complex (Supplementary Fig. 5). Similar to the wild-type complex, the Smc5/6 ATPase mutant dynamically associated with linear ssDNA and dsDNA regions and accumulated at DNA junctions (Supplementary Fig. 6). Further, ATP still enhanced the mutant complex's interaction with dsDNA but not with junction DNA, similar to the wild-type behavior (Supplementary Fig. 6e, Supplementary Movie 2). The most likely explanation for the common behavior of the two complexes is that ATP binding, but not hydrolysis, contributes to optimal Smc5/6 association with dsDNA.

## Nse1-3-4 aids Smc5/6 association with dsDNA and assembly on junction DNA
Our results thus far indicated that Smc5/6 has a basal ATP-independent dsDNA association activity, which could be modified by ATP binding.

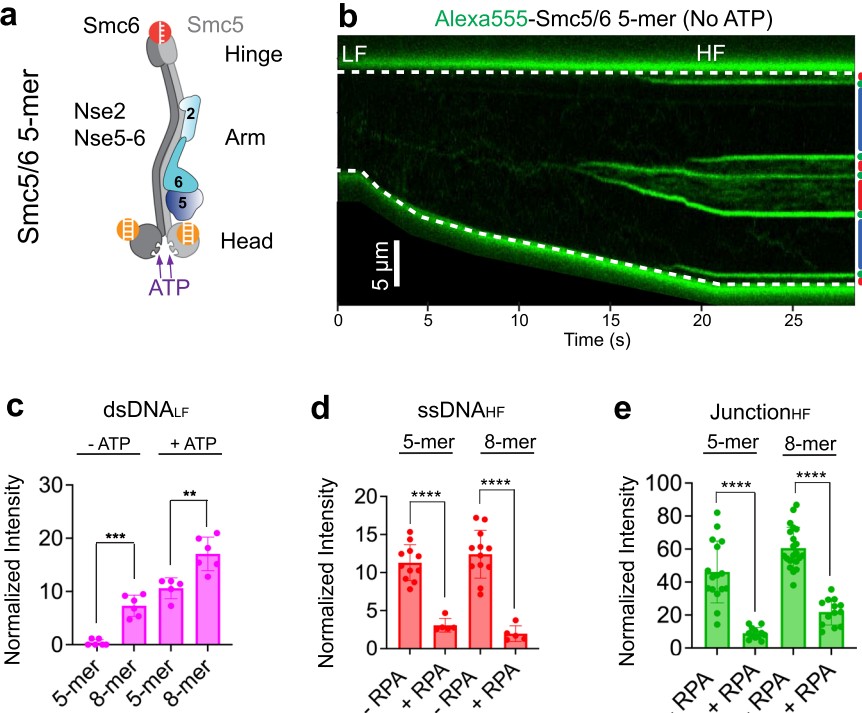

**Fig. 3 | Nse1-3-4 subcomplex contributes to Smc5/6's association with DNA.**
**a** Schematic of the Smc5/6 5-mer complex, which lacks the Nse1-3-4 subcomplex.
**b** Representative kymograph of a λ DNA tether being stretched from LF to HF in the presence of 20 nM Alexa555-Smc5/6 5-mer (green) without ATP. dsDNA (blue), ssDNA (red), and junction DNA (green) regions of the tether are indicated to the right of the kymograph. **c** Fluorescence signals of Smc5/6 5-mer and 8-mer complexes on dsDNA under LF without or with ATP (no RPA). Bar heights indicate the group mean and error bars represent standard deviation. $P$ values were determined from two-tailed unpaired $t$-tests with Welch's correction (**$P < 0.01$; ***$P < 0.001$). $P = 0.0002$ for −ATP condition, and $P = 0.0028$ for +ATP condition. Sample sizes are: 5-mer dsDNA$_{LF}$ −ATP ($n = 6$); 8-mer dsDNA$_{LF}$ −ATP ($n = 6$); 5-mer dsDNA$_{LF}$ + ATP ($n = 5$); and 8-mer dsDNA$_{LF}$ + ATP ($n = 6$), where $n$ indicates the number of regions analyzed. **d** Fluorescence signals of Smc5/6 5-mer and 8-mer complexes on ssDNA

under HF in the presence of ATP without or with RPA. Bar heights indicate the group mean and error bars represent standard deviation. $P$ values were determined from two-tailed unpaired $t$-tests with Welch's correction (****$P < 0.0001$). Sample sizes are: 5-mer ssDNA$_{HF}$ −RPA ($n = 11$); 5-mer ssDNA$_{HF}$ + RPA ($n = 6$); 8-mer ssDNA$_{HF}$ −RPA ($n = 12$); 8-mer ssDNA$_{HF}$ + RPA ($n = 5$), where $n$ indicates the number of regions analyzed. **e** Fluorescence signals of Smc5/6 5-mer and 8-mer complexes on junction DNA under HF in the presence of ATP without or with RPA. Bar heights indicate the group mean and error bars represent standard deviation. $P$ values were determined from two-tailed unpaired $t$-tests with Welch's correction (****$P < 0.0001$). Sample sizes are: 5-mer Junction$_{HF}$ −RPA ($n = 16$); 5-mer Junction$_{HF}$ + RPA ($n = 12$); 8-mer Junction$_{HF}$ −RPA ($n = 21$); and 8-mer Junction$_{HF}$ + RPA ($n = 13$), where $n$ indicates the number of regions analyzed. Source data for panels **c**, **d**, and **e** are provided within the Source Data file.

As mentioned, the ATP-dependent effect is likely mediated by Smc5/6 head dimerization upon ATP binding. For ATP-independent dsDNA binding, we considered the role of the dsDNA binding subcomplex Nse1-3-4, since Nse3 binding to the Smc5/6 head region in the absence of ATP could in principle provide another dsDNA binding mode (Supplementary Fig. 1b)[9,17]. To this end, we tested Smc5/6 complexes that lacked Nse1-3-4, referred to as the Smc5/6 5-mer complex hereafter (Fig. 3a, Supplementary Fig. 2a). We found that the Smc5/6 5-mer largely lacks both ATP-mediated DNA entrapment and ATPase activities (Supplementary Fig. 2c, e), consistent with previous observation that Nse1-3-4 contributes to ATP-dependent DNA clamp formation and ATP hydrolysis[7,18].

Compared with the Smc5/6 8-mer, the Smc5/6 5-mer showed reduced dsDNA binding regardless of the presence of ATP (Fig. 3b, c), indicating that Nse1-3-4 indeed contributes to both ATP-dependent and -independent Smc5/6 binding to dsDNA. The addition of ATP increased Smc5/6 5-mer intensities on dsDNA, although still lower than those of the Smc5/6 8-mer (Fig. 3c, Supplementary Fig. 7a).

On the other hand, the Smc5/6 5-mer was proficient for binding to stretched ssDNA regions (Fig. 3b, d), suggesting that ssDNA binding per se does not require Nse1-3-4. This could be rationalized by the spatial separation between the hinge region and the Nse1-3-4 subcomplex located close to the head region. Interestingly, compared with the Smc5/6 8-mer, the Smc5/6 5-mer exhibited reduced streak signals at junction DNA, particularly in the presence of RPA (Fig. 3e,

Supplementary Fig. 7b, c). This result provides evidence for an additional role of Nse1-3-4 in aiding Smc5/6 assembly at DNA junctions (see Discussion).

## Smc5/6 stably binds to synthetic replication forks

Our data thus far revealed multifaceted behaviors of Smc5/6 on DNA: it dynamically interacts with linear dsDNA while stably associating with and accumulating at junction DNA. Smc5/6 signal accumulation could occur either exclusively at the ss-dsDNA junction site or along the adjacent flap of free ssDNA emanating from the junction site. We went on to distinguish these two possibilities and tested whether Smc5/6 could stably associate with and/or accumulate on a DNA junction mimicking a bona fide replication fork without a ssDNA flap. To this end, we constructed a three-armed DNA fork substrate via multi-step ligation (Fig. 4a, Supplementary Fig. 8a). This construct contained short ssDNA regions on the leading and lagging strand arms (24 nt and 100 nt, respectively). The forked DNA was tethered between two beads through a biotinylated lagging strand arm and the parental strand arm, yielding a 64.5-kbp-long tether, and was examined at the low force regime. The predetermined fork position in this tether allowed unambiguous identification of the DNA junction site. To validate successful assembly of this tether, we incubated it with LD555-labelled eukaryotic replicative helicase complex CMG, which is known to reside at the fork junction[14], along with its cofactor Mcm10 and ATP. Indeed, we observed stable CMG fluorescent signals specifically at the

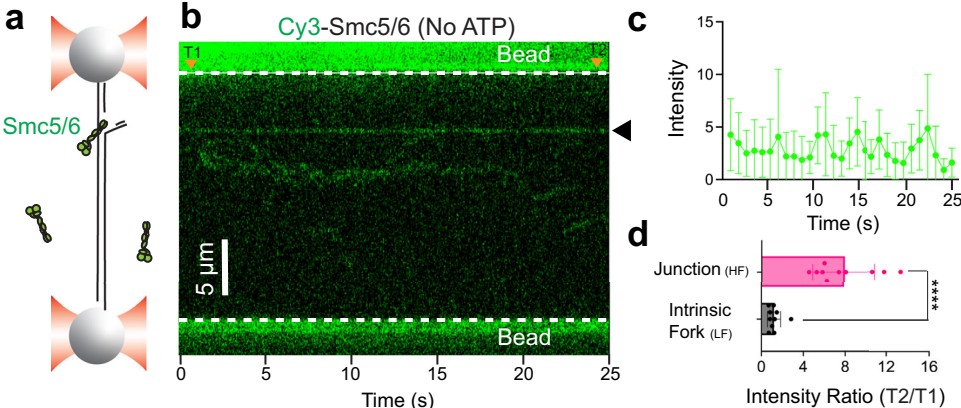

**Fig. 4 | Smc5/6 stably binds to replication forks. a** Schematic showing a forked DNA tethered between two beads under LF. The intrinsic fork is located one quarter distance of the total tether length from the proximal bead.
**b** Representative kymograph of a forked DNA tether in the presence of 5 nM Cy3-Smc5/6 (green). Stable binding of Smc5/6 at the intrinsic fork position was observed on 10 out of 22 (45%) tethers tested. **c** Quantification of the Smc5/6 fluorescence signals at the intrinsic fork (marked by the black arrow in panel **b**) over time. Data points indicate the averaged photon count per frame ($n = 10$ frames) at the fork site and error bars represent standard deviation. **d** Smc5/6 fluorescence intensity ratio between two timepoints (T1 and T2 as indicated by orange arrows in panel **b**) at an intrinsic fork under LF versus at a DNA junction generated at HF. Bar heights indicate the group mean and error bars represent standard deviation. *P* value was determined from a two-tailed unpaired *t*-test with Welch's correction (****$P < 0.0001$). The sample sizes are: Junction$_{HF}$ ($n = 10$) and Intrinsic Fork$_{LF}$ ($n = 10$), where $n$ indicates the number of Smc5/6 streaks analyzed. Source data for panels **c** and **d** are provided within the Source Data file.

expected position of the fork junction located 16 kb from the bead attached to the lagging strand arm (Supplementary Fig. b–c).

Next, we incubated this DNA construct with Cy3-labelled Smc5/6 complexes and examined Smc5/6 binding as we did for CMG. Significantly, stable Smc5/6 association was observed specifically at the fork junction and the interaction did not require ATP (Fig. 4b). Both results were consistent with those derived from our DNA stretching experiments. We thus concluded that Smc5/6 could stably associate with junction DNA regardless of the force regime. A difference between these two conditions was that only the force-induced fork junctions contained long ssDNA flaps and showed Smc5/6 signal accumulation over time (Figs. 4c vs. 1d). The intrinsic fork substrate containing short ssDNA segments near the junction did not support Smc5/6 accumulation, as the fluorescent signals stayed constant (Fig. 4c, d). Collectively, these results suggested that Smc5/6 stably binds to DNA fork junctions and accumulates when long stretches of free ssDNA are available.

To estimate the number of Smc5/6 complexes at both types of junction sites, we took advantage of stepwise fluorescence decreases over an extended observation window presumably due to the stochastic photobleaching of single fluorophores (Supplementary Fig. 3d). Based on the magnitude of signal decrease for each step, which reflected the fluorescence intensity from a single Cy3-Smc5/6 complex, we estimated that one Smc5/6 complex resided at the intrinsic replication fork junction (Fig. 4c), while up to 25 complexes accumulated on force-induced junctions, likely by occupying the adjacent free ssDNA (Figs. 1d, 2c).

## Smc5/6 accumulates on free ssDNA
To further test the notion that Smc5/6 assembles onto free ssDNA emanating from ss-dsDNA junctions generated by DNA stretching, we utilized the *E. coli* SSB protein, which, unlike RPA, wraps around free ssDNA but not tethered ssDNA under tension[15]. The λ DNA tether was incubated with both LD650-labelled SSB and Cy3-labelled Smc5/6 8-mer complex. As the inter-bead distance increased, SSB signals first emerged (Fig. 5a, Supplementary Fig. 9), indicating the rapid formation of SSB-ssDNA complexes[19]. Subsequently, Smc5/6 signals that overlapped with those of SSB appeared and grew stronger over time (Fig. 5b, Supplementary Movie 3). We found that Smc5/6 accumulation on SSB-bound ssDNA did not require ATP.

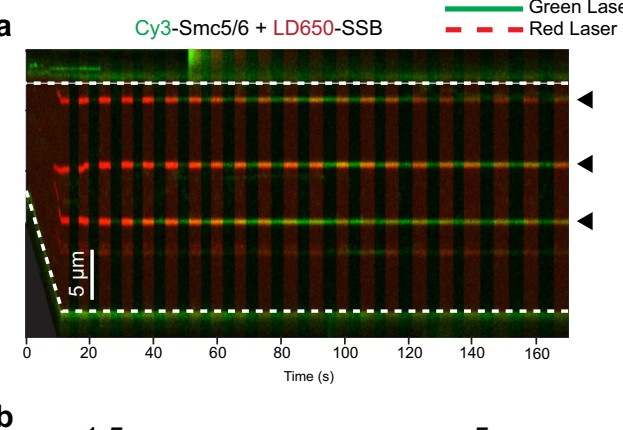

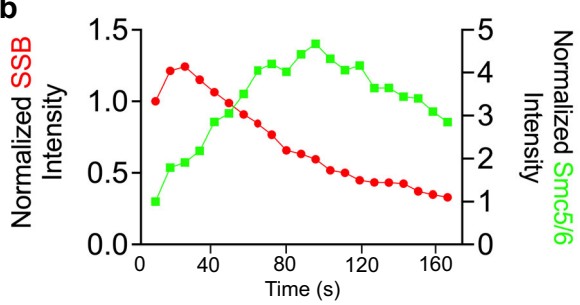

**Fig. 5 | Smc5/6 assembles on free ssDNA near junctions. a** Representative kymograph of a λ DNA tether being stretched from LF to HF in the presence of 20 nM Cy3-Smc5/6 (green) and 10 nM LD650-SSB (red) without ATP. The green laser was on for the entire duration of the kymograph whereas the red laser alternated between on and off. **b** Quantification of the SSB and Smc5/6 fluorescence signals (normalized to the first timepoint) averaged from 18 junction DNA sites such as those indicated by the black arrows in panel **a**. Source data are provided within the Source Data file.

## Smc5/6 suppresses ssDNA annealing at DNA junctions
The biological implications of the dichotomous behavior of Smc5/6 in binding linear dsDNA versus junction DNA could be manifold (see Discussion). Here we specifically asked whether such properties could help stabilize DNA junction structures, which is relevant to Smc5/6-

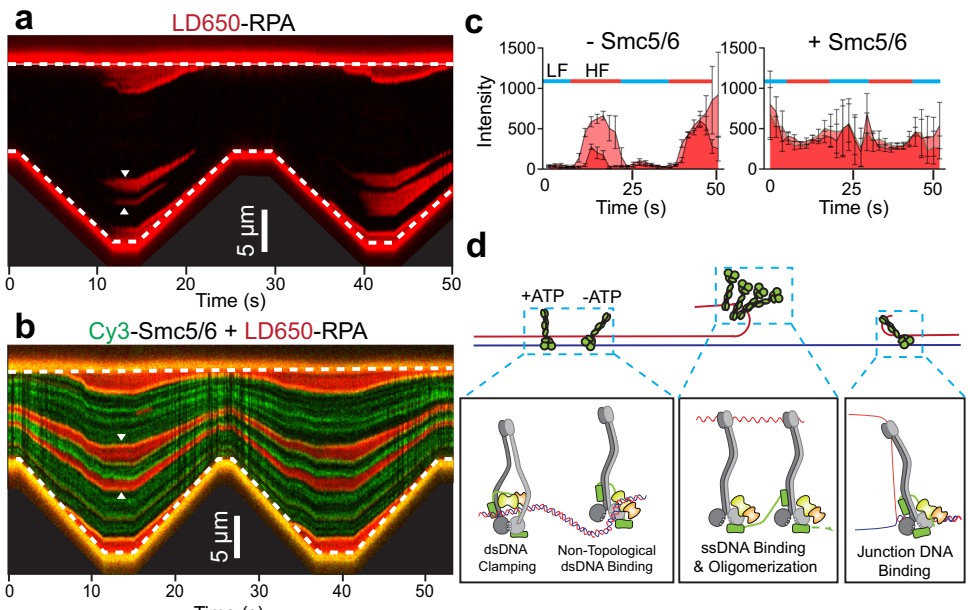

**Fig. 6 | Smc5/6 stabilizes DNA junctions. a** Representative kymograph of a λ DNA tether alternating between LF and HF in the presence of 10 nM LD650-RPA (red). RPA was ejected from the DNA in the transition from HF to LF (examples indicated by white arrows) due to the re-annealing of complementary ssDNA.
**b** Representative kymograph of a λ DNA tether alternating between LF and HF in the presence of 10 nM LD650-RPA (red), 20 nM Cy3-Smc5/6 (green), and 2 mM ATP. The RPA signals remained on the DNA throughout pulling-relaxation cycles (examples indicated by white arrows). **c** Quantification of the RPA fluorescence signals (from the individual streaks indicated by the white arrows in panels **a** and **b**) over time in the absence or presence of Smc5/6. LF and HF regimes are indicated by blue and red bars, respectively. Data points indicate the averaged photon count per frame (n = 10 frames) of a region along the RPA streak and error bars represent standard deviation. Source data are provided within the Source Data file. **d** (*Top*) Summary of the different Smc5/6 binding behaviors on DNA observed in our study. (*Bottom*) Working model for Smc5/6 binding modes on different types of DNA.

based replication fork protection seen in vivo[2]. We have shown previously that RPA coating the ssDNA regions generated by force-induced peeling could be evicted when the tether was relaxed due to re-annealing of the ssDNA[14]. We performed pulling-relaxation cycles in the presence of LD650-labelled RPA and reproduced these results (Fig. 6a, Supplementary Fig. 10a). Remarkably, the addition of Cy3-Smc5/6 prevented ssDNA re-annealing as evidenced by persistent RPA association with ssDNA even after the tether was fully relaxed (Fig. 6b–c, Supplementary Fig. 10b). Given Smc5/6's DNA binding behaviors described above, this effect could be mediated by the stable binding of Smc5/6 at ss-dsDNA junctions as well as its sequestration of nearby free ssDNA, which creates an energy barrier to prevent the annealing of complementary ssDNA strands.

## Discussion

The three eukaryotic SMC complexes play different roles in genome organization and maintenance. Previous studies have suggested that Smc5/6 influences multiple processes required for genomic stability and some of its roles pertain to the management of impaired replication forks and joint DNA structures generated during DNA repair[2]. Both types of DNA substrates contain junction DNA consisting of ssDNA adjacent to dsDNA regions. How Smc5/6 interacts and influences these junction DNA substrates versus dsDNA and ssDNA has been unclear. In this work, we addressed this outstanding question by examining fluorescently labelled Smc5/6 on linear dsDNA, ssDNA, and junction DNA that co-existed on the same substrate. Our data revealed dynamic association of Smc5/6 with linear dsDNA but stable binding to junction DNA. The complex also showed a striking capacity to accumulate along free ssDNA even in the presence of high-affinity ssDNA binding proteins. Mechanistically, we found that ATP binding by the Smc5-6 head regions and the Nse1-3-4 subcomplex both contribute to dsDNA association and that Nse1-3-4 additionally aids Smc5/6 accumulation on ssDNA. To our knowledge, the multifaceted binding behavior

observed here for Smc5/6 has not been reported for cohesin, condensin, or bacterial SMCs. These unique properties of Smc5/6 likely contribute to its diverse roles in genome protection and viral DNA restriction.

### Smc5/6 dynamically interacts with dsDNA in two modes

Our data suggest that Smc5/6 interacts with dsDNA in both ATP-dependent and ATP-independent modes. The ATP dependence seen here likely reflects a requirement of ATP binding rather than hydrolysis since the ATPase mutant of Smc5/6 did not reduce dsDNA association. This conclusion also fits with biochemical and structural data showing that ATP binding by Smc5-6 favors dsDNA entrapment and that the ATP-bound Smc5-6 head regions and Nse1-3-4 form a clamp that can enclose a DNA duplex (Supplementary Fig. 1a)[4,7,16]. Combining these data with our observation that ATP stimulates topological entrapment of dsDNA by Smc5/6, we infer that the ATP-dependent mode of dsDNA binding by Smc5/6 could be in the form of 'dsDNA clamping' (Fig. 6d, left). However, we cannot rule out other possible forms given the paucity of structural data currently available. Future studies using mutants that affect the DNA clamp formation will provide more insights on this binding mode.

Structural details of ATP-independent dsDNA binding by Smc5/6 remain to be revealed. Our data suggest that this mode of dsDNA binding requires the Nse1-3-4 subcomplex, which contains the major DNA binding module Nse3[7,17]. Interestingly, a recently published ATP- and DNA-free structure showed that Nse3 binds to the Smc5 head region[9] (Supplementary Fig. 1b). This configuration in principle could support ATP-independent dsDNA association since the adjacent Nse3 and Smc5 head region possess over a dozen DNA binding residues that could allow dsDNA docking over these regions. This hypothetical 'non-topological dsDNA binding' mode (Fig. 6d, left) does not offer the same level of stability of DNA association as the 'dsDNA clamping' mode. As such, the two modes of interaction could meet the different

needs of Smc5/6 during various processes of genome management. This is further discussed below.

## Smc5/6 stably associates with junction DNA and accumulates on free ssDNA

Using a three-armed replication fork substrate, we found that Smc5/6 stably binds to the DNA junction in an ATP-independent manner. Since the hinge region is the only part of Smc5/6 known to bind ssDNA[8], we infer that it may bind ssDNA located at fork junctions regardless of ATP. Testing this notion will require determination of the ssDNA binding residues of the budding yeast Smc5/6 in the future. Extending from this model, one could envision that Smc5/6 engagement with a DNA junction entails simultaneous binding of the hinge region with ssDNA and the head region with the adjacent dsDNA of a junction, which should confer more stability than binding to ssDNA or dsDNA alone (Fig. 6d, right). As such, Nse1-3-4 may contribute to ss-dsDNA junction association by forming an Nse3 and Smc5 binding surface that facilitates dsDNA binding. While this interpretation is consistent with the current knowledge of how Smc5/6 interacts with ssDNA and dsDNA and can explain our data, its validation requires future mutagenesis and structural studies.

We found little evidence in our kymographs that Smc5/6 must slide on dsDNA prior to accumulation at ss-dsDNA junctions, thus dsDNA sliding is not a prerequisite for junction binding; rather our data favor the idea that Smc5/6 can engage directly with junction DNA. A major difference of Smc5/6 binding to DNA junctions when using stretched DNA versus a replication fork structure was the Smc5/6 assembly on free ssDNA on the former substrate. The observed ssDNA-based Smc5/6 assembly was initially blocked by SSB wrapping the free ssDNA, but at later time points, Smc5/6 still assembled onto SSB-occupied ssDNA. One logical explanation of Smc5/6 accumulation on ssDNA may be complex oligomerization (Fig. 6d, middle). We speculate that oligomerization could be mediated by tethering through the elongated Nse4. It is known that the two Nse4 terminal domains can engage the Smc5 head and the Smc6 coiled-coil regions in cis[7]. The same interaction mechanisms may also be able to bridge two complexes together wherein the two terminal regions of Nse4 bind to Smc5 and Smc6 in adjacent complexes. Testing this notion in the future will further our understanding of Smc5/6.

We observed that Smc5/6 can stabilize DNA junctions and disfavor ssDNA re-annealing. As ssDNA binding by RPA alone could not achieve these effects in our experimental setup, a logical interpretation is that the observed effects by Smc5/6 are due to its multifaceted capacities of binding to different forms of DNA. We envision that the stable association of Smc5/6 with fork junctions as well as Smc5/6 assembly on free ssDNA provide sufficient free energy to disfavor the re-annealing of two complementary ssDNA strands.

## Implications of Smc5/6's multifaceted DNA binding capacities

Our work reveals multiple DNA interaction modes by Smc5/6. Taking consideration of available structural and biochemical data, we speculate that these modes are enabled by distinct complex conformations and support different functionalities of Smc5/6 (Fig. 6d). For example, ATP-dependent 'dsDNA clamping' may help Smc5/6 translocate on dsDNA or facilitate its recently reported DNA compaction and loop extrusion activities[5,6,20]. On the other hand, the ATP-independent 'non-topological dsDNA binding' mode on its own may facilitate Smc5/6 sampling along dsDNA without stable association. If dsDNA is adjacent to a region of ssDNA, the Smc5/6 hinge domain may engage with the ssDNA and transition the complex into a 'junction binding' mode.

The stable association between Smc5/6 and DNA junction structures could allow the complex to execute specific roles at damaged replication forks or recombination intermediates. For example, stable interaction with DNA junctions by Smc5/6 may provide a physical barrier against replication fork reversal enzymes, an in vivo function

suggested by previous work[21]. Also, the assembly of multiple Smc5/6 complexes on free ssDNA may underlie the reported protective effects of Smc5/6 on DNA breaks and the prevention of spurious ssDNA invasion[22–24]. Finally, the collective action of Smc5/6 in stable association with junction DNA and ssDNA emanated from the junction could disfavor ssDNA annealing, thus stabilizing branched DNA structures such as those formed during DNA replication and repair. Future research that tests these possibilities will enhance our understanding of how Smc5/6 engages and manipulates DNA in different genomic contexts.

## Methods

### Smc5/6 protein purification

The different subunits of the *S. cerevisiae* Smc5/6 were codon optimized and synthesized, and then cloned into bidirectional galactose inducible promoter vectors using Gibson assembly. Then the vectors were linearized and integrated into W303-1a strain to generate the desired complex expression strains. Cells hosting the desired complex were grown at 30 °C in YP-GL (YP + 2% glycerol/2% lactic acid) medium to an OD600 of 0.8-1.0, and proteins were induced by adding 2% galactose for 4 h. Cells were harvested by centrifugation and washed with 1 M sorbitol/25 mM HEPES-KOH pH 7.6, buffer E (45 mM HEPES-KOH pH 7.6, 10% glycerol, 0.02%NP40)/100 mM NaCl, then resuspended in 0.5 volumes of buffer E/100 mM NaCl/1 mM DTT/1x protease inhibitor cocktail (Sigma)/1x cOmplete Ultra EDTA-free protease inhibitor (Roche), and frozen dropwise in liquid nitrogen. The resulting popcorn was crushed in a freezer mill (SPEX CertiPrep 6850 Freezer/Mill) for 7 cycles of 2 min at a rate of 15 impacts per second. Crushed cell powder was thawed on ice and resuspended with 1 volume of buffer E/100 mM NaCl/1 mM DTT. Extra 200 mM NaCl was added, and extracts were centrifuged in a T647.5 rotor (Thermo Scientific) for 30 min at 177,469 x g (40,000 rpm), 4 °C. The supernatant was supplemented with 2 mM $CaCl_2$ and incubated with calmodulin affinity resin for 2 h at 4 °C. The calmodulin affinity resin was washed by 10 CVs of buffer E/300 mM NaCl/2 mM $CaCl_2$/1 mM DTT, and the protein was eluted by 10 CVs of buffer E/300 mM NaCl/1 mM EDTA/2 mM EGTA/1 mM DTT. Peak fractions were pooled and fractionated by gel-filtration on a Superose 6 Increase column equilibrated in buffer 45 mM HEPES-KOH pH 7.6, 10% glycerol, 200 mM NaCl, 0.02% NP40, 1 mM DTT.

### Protein labeling

**Site-specific labeling.** Site-specific labeling of Smc5/6 and CMG used SFP synthase (4'-phosphopantetheinyl transferase), which specifically recognizes a short peptide tag and catalyzes the covalent transfer of CoA-functionalized moieties to a single serine residue within the tag via a phosphopantetheinyl linker. To obtain Smc5/6 labelled with a single Cy3, we generated an Smc5/6 complex with an inserted "S6" peptide (GDSLSWLLRLLN) on the Smc5 C-terminus. S6-tagged Smc5/6, SFP, and Cy3-CoA were incubated at a 1:3:5 molar ratio overnight at 4 °C in the presence of 10 mM $MgCl_2$. Excess dye and SFP were removed by buffer exchange on the Amicon Ultra-0.5 Centrifugal Filter Unit (Millipore) with a storage buffer containing 45 mM HEPES-KOH pH 7.6, 200 mM NaCl, 10% glycerol, and 1 mM DTT. The final Cy3-Smc5/6 complex was aliquoted, flash frozen, and stored at −80 °C. LD555-CMG was generated with the same method as Cy3-CMG in a previous publication[14], in which an S6 sequence was inserted in the C-terminus of Cdc45 and the purified protein was labelled at a 1:2:5 molar ratio of CMG-S6, SFP, and LD555-CoA for 1 h in room temperature in the presence of 10 mM $MgCl_2$.

**Non-specific labeling.** We used AlexaFluor555 (Alexa555) NHS ester (Thermo Fisher; Catalog # A20009) to non-specifically label the primary amines of Smc5/6 subunits. Preferential N-terminal labeling was achieved by labeling at low pH (7.0) for an NHS ester reaction. Labeling was performed with a labeling buffer containing 45 mM HEPES-KOH pH 7.0, 200 mM NaCl, 1 mM DTT, and 0.25 mM EDTA pH 8.0.

The Smc5/6 sample was incubated with Alexa555 dye at a 1:5 molar ratio for 1 h at room temperature in the dark. The reaction was quenched with 30 mM Tris-HCl pH 7.0 for 5 min at room temperature. Excess dye was removed from Smc5/6 by buffer exchange on the Amicon Ultra-4 Centrifugal Filter (Millipore) with a storage buffer containing 45 mM HEPES-KOH pH 7.6, 200 mM NaCl, 10% glycerol, and 1 mM DTT. The final labelling ratio was estimated to be >80% for all Smc5/6 samples. The labelled Smc5/6 complex was aliquoted, flash frozen, and stored at −80 °C. LD650-RPA and LD650-SSB were labelled using a similar protocol, but with a labelling buffer containing 50 mM HEPES-KOH pH 7.0, 150 mM NaCl, 1 mM DTT, and 0.25 mM EDTA, and a storage buffer containing 20% glycerol, 30 mM HEPES pH 7.9, 150 mM NaCl, 1 mM DTT, and 0.25 mM EDTA as previously described[14].

## DNA substrate preparation

**Biotinylated λ DNA.** To create a terminally biotinylated dsDNA template, the 12-base 5′ overhang on each end of the genomic DNA from bacteriophage λ (48,502 bp, Roche) was filled in with a mixture of unmodified and biotinylated nucleotides by the exonuclease-deficient DNA polymerase I Klenow fragment (New England BioLabs). Reaction was conducted by incubating 10 nM λ DNA, 33 µM each of dGTP/dATP/biotin-11-dUTP/biotin-14-dCTP (Thermo Fisher) and 5 U Klenow in 1x NEB2 buffer at 37 °C for 45 min, followed by heat inactivation for 20 min at 75 °C. DNA was then ethanol precipitated overnight at −20 °C in 2.5x volume cold ethanol and 300 mM sodium acetate pH 5.2. Precipitated DNA was recovered by centrifugation at 20,000 x *g* for 15 min at 4 °C. After removing the supernatant, the pellet was air-dried, resuspended in TE buffer (10 mM Tris-HCl pH 8.0, 1 mM EDTA) and stored at 4 °C.

**Three-armed fork substrate.** To prepare the replication fork DNA substrate, λ DNA (New England BioLabs) was first biotinylated at one end by annealing a 41-mer oligonucleotide and a 29-mer with 5′ dual biotin via the cos overhang and ligating with T4 DNA Ligase (New England BioLabs) as previously reported[12]. The leading strand arm was attached by annealing and ligating a 90-mer oligonucleotide to λ DNA via the second cos site (5′ phosphate-GGGCGGCGACCTCTAG CGTGG GTAGGGACTTACTGAGGATAGGTTTTTTTTTTGAGGCAAGAAAGAAG GAAGTGCTCTATGAGACGGGAA). The lagging strand arm was made from purified pRGEB32 (ref. [25]). The 16 kb plasmid was linearized by restriction digest with BsaI-HF v2 (New England BioLabs) and biotinylated at one end by annealing and ligating oligonucleotides 1 and 2 (oligo 1: 5′ dual-biotin TTTAGT^biotin CCT^biotin CAAAGCCTCTGTAGC; oligo 2: 5′ phosphate-AAACGCTACAGAGGCTTTGAGGACTAAA). Oligo 1 contained a 5′ dual-biotin modification and two internal biotin-dTs, as noted in the sequence above. The opposite end was then annealed and ligated to λ DNA via a 155-mer oligonucleotide containing a 100x dT region. Finally, the leading strand arm overhang was primed by annealing a 23-mer oligonucleotide (TTCCCGTCTCATAGAGCACT TCC) leaving a 24-nt single-stranded gap (gap sequence: TTTTTTT TTTGAGGCAAGAAAGAA) before the fork junction. The substrate was purified and stored in TE buffer at 4 °C. All synthetic oligonucleotides were ordered from Integrated DNA Technologies. pRGEB32 was a gift from Yinong Yang (Addgene plasmid # 63142).

## Single-molecule experiments

**Data acquisition.** Single-molecule experiments were performed at room temperature on a LUMICKS C-Trap instrument combining three-color confocal fluorescence microscopy with dual-trap optical tweezers. A computer-controlled stage enabled movement of the optical traps within a microfluidic flow cell (Supplementary Fig. 1c). Laminar flow separated channels 1–3, which were used to form DNA tethers between 4.89-µm streptavidin-coated polystyrene beads (Spherotech). Under constant flow, a single bead was caught in each trap in channel 1. The traps were then moved to channel 2 containing

the biotinylated DNA of interest. By moving one trap back and forth along the axis of the flow direction, a DNA tether could be formed and detected via a change in the force-extension curve. The traps were then moved to channel 3 containing only buffer (50 mM potassium glutamate, 50 mM Tris-HCl pH 7.5, 2.5 mM MgCl2, and 1 mM DTT), and the presence of a single DNA tether was verified by the force-extension curve. Orthogonal channels 4 and 5 served as the experimental and imaging channels that contained proteins of interest. Flow was turned off during data acquisition. Force data were collected at 50 kHz and 15 Hz. Cy3, Alexa555, and LD555 fluorophores were excited by the 532-nm laser line (green laser), and the LD650 fluorophore was excited by the 638-nm laser line (red laser). The green laser operated at ~2.5 µW and the red laser at ~4.2 µW measured at the objective. Kymographs were generated via a confocal line scan through the center of the two beads, with a line time of approximately 45 milliseconds and a pixel time of 0.1 milliseconds. To visualize protein behavior on the λ DNA tether under a low-force regime, the inter-bead distance was kept at ~10 µm. To visualize protein behavior under a high-force regime, the inter-bead distance was increased to 19–22 µm. For experiments involving repeated stretching-relaxation cycles, the inter-bead distance alternated between LF and HF regimes at a constant velocity of ~1 µm s⁻¹. To test topological interaction of Smc5/6 with dsDNA, we utilized a high-salt buffer containing 500 mM NaCl, 50 mM Tris-HCl pH 7.5, 2.5 mM MgCl2, and 1 mM DTT.

**Data analysis.** Kymographs were processed and visualized using custom software written by John Watters (https://harbor.lumicks.com/single-script/c5b103a4-0804-4b06-95d3-20a08d65768f), which incorporates tools from the lumicks.pylake Python library and other Python modules (Numpy, Matplotlib, Pandas). To create the Supplemental Movies, continuous 2D scans were collated together by ImageJ (Version 1.53). To quantify fluorescence signals, the absolute photon count per pixel for each classified DNA region was averaged over 20 frames to create a datapoint. Background fluorescence was calculated by averaging the absolute photon count over ~1000 pixels in the background space of the same kymograph and subtracted from the corresponding datapoints. To track fluorescence growth over time, discrete signal streaks in a kymograph were identified using custom software Area Photon Count Extractor (https://harbor.lumicks.com/single-script/23f33367-7b02-4762-9457-37348da59194). A series of pixel boxes (a column of 5 pixels over 10 frames) was created along a streak of interest. Each datapoint is an averaged photon count per frame of a pixel box (background subtracted). Graphs and plots were created using Prism 9 version 9.3.1.

## ATPase assays

The ATPase activity was assayed with the EnzChek Pyrophosphate Assay Kit (E-6645). The inorganic phosphate formed during the reactions was measured at 37 °C as specified in the manual in the buffer including purine nucleoside phosphorylase (PNP), 2-amino-6-mercapto-7-methylpurine ribonucleoside (MESG) and supplemented with 2 mM ATP and 250 nM Smc5/6 complex. For the DNA stimulation tests, 2 µM 72-bp dsDNA was added to the reactions. The signal was monitored for 2 h by measuring absorbance at 360 nm in an Epoch 2 Microplate Spectrophotometer.

## DNA entrapment assays

A 100-µL reaction containing 300 nM Smc5/6 complex and 2.5 µg plasmid (pUC19, 2686 bp, either circular or EcoRI linearized) was mixed in DNA binding buffer (10 mM HEPES-KOH pH 7.5, 150 mM KAc, 2 mM MgCl2, 20% glycerol, 2 mM CaCl2) with or without 2 mM ATP. After 30 min of incubation at room temperature, 500 µL of ice-cold high-salt buffer (20 mM Tris pH 7.5, 1 M NaCl, 2 mM CaCl2) was added and mixed well, then incubated with 20 µL pre-washed calmodulin affinity resin at 4 °C for 1 h. In the controls, 500 µL DNA binding buffer

was used instead. The resin was transferred into Micro Bio-Spin Columns (Bio-Rad) and washed by 500 μL high-salt buffer three times. In the controls, 500 μL DNA binding buffer was used instead. Bead-bound materials were eluted with 50 μL elution buffer (20 mM Tris pH 7.5, 250 mM NaCl, 2 mM EDTA, 2 mM EGTA). Aliquots of the elute were mixed with 6x SDS-containing loading dye, heated at 65 °C for 10 min before loading onto a 1% agarose gel to examine recovered DNA.

### Yeast genetics
Strain construction and tetrad analyses were completed according to standard protocols. Genotype of spores after tetrad dissection was done based on the marker linked with each indicated tagged alleles.

### Quantification and statistical analysis
Statistical tests and sample sizes are reported in the figure legends. Representative kymographs are shown in the figures.

### Reporting summary
Further information on research design is available in the Nature Portfolio Reporting Summary linked to this article.

## Data availability
Source data are provided with this paper as a Source Data file. Additional data are available from the corresponding authors upon reasonable request. Source data are provided with this paper.

## Code availability
All specified scripts used to process and analyze C-trap experiments can be accessed on the LUMICKS Harbor site (https://harbor.lumicks.com).

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

## Acknowledgements

We thank Drs. Zhi Qi and Hengyao Huang for their help during the initial stages of this work, Danying Guan and Gabriella Chua for technical support, John Watters for single-molecule data analysis, Sai Li and Michael Wasserman for reagents. Je.T.C was supported by an NCI F30 fellowship under award F30CA275379 and a Medical Scientist Training Program grant from NIGMS under award T32GM007739 to the Weill Cornell/Rockefeller/Sloan Kettering Tri-Institutional MD/PhD Program. M.E.O. was supported by R01GM115809 and Howard Hughes Medical Institute. X.Z. was supported by NIGMS grant R35GM145260 and a Center for Epigenetics Research grant. S.Liu was supported by the Robertson Foundation, the Alfred P. Sloan Foundation, the Pershing Square Sohn Cancer Research Alliance, and an NIH Director's New Innovator Award under award DP2HG010510.

## Author contributions

X.Z. and S.Liu oversaw the project. Je.T.C. performed single-molecule experiments and analyzed the data. S.Li prepared protein constructs and performed biochemical and genetic experiments. E.C.B., Jo.C. and M.E.O. contributed to reagents. T.T. and C.H. contributed to protein purification and characterization. Je.T.C., S.Li, X.Z. and S.Liu wrote the paper with inputs from other authors.

## Competing interests
The authors declare no competing interests.
