## [Peer Review File · Nature Communications]

Smc5/6's multifaceted DNA binding capacities stabilize branched DNA structuresREVIEWER COMMENTS

Reviewer #1 (Remarks to the Author):

The manuscript describes a detailed analysis of Smc5/6 binding to different types of DNA substrates (dsDNA, ssDNA and ssDNA/dsDNA junctions) at different tensions and unravels an intriguing ability of this complex to displace SSB and prevent strand reannealing. Despite extensive studies of Smc5/6 and its critical roles in a number of key cellular processes, including DNA repair and replication, the mechanisms underlying the functions of Smc5/6 remain elusive. The new single-molecule data presented here are therefore not only of outstanding quality but also extremely timely, and they substantially further our understanding of this crucial complex. I thus strongly recommend the publication of this study at Nature Communications, and, given the competitive nature of this 'hot field', as soon as possible.

I do not have major points that must be addressed, but the manuscript would benefit from the authors addressing the following minor points/questions:

The authors should include a more detailed discussion of the results they obtained in relation to the demonstration of loop extrusion and DNA translocation activities for Smc5/6 in a recent preprint (ref. 20). For example, it would be helpful for readers if the authors could comment on the choice of the low-force regime (around 5 pN) compared to a force regime that permits translocation and loop extrusion (<1 pN).

There seems to be an apparent contradiction between the circular DNA entrapment results, the statement that "ATP enhancement of Smc5/6 binding to dsDNA seen here at the single-molecule level is consistent with biochemical data showing that ATP promotes...topological entrapment of circular dsDNA" (line 174) and the statement that "... signal at linear dsDNA... rapidly dropped to background levels" (line 156). It seems like a dsDNA substrate tethered between two microspheres should not permit a ring to slide off the ends, just like circular DNA, and therefore topological binding is expected to last very long. For that reason, the fast dissociation of Smc5/6-dsDNA complex seems to argue against topological entrapment. The data in Fig. S3A seem to feature many long-lasting Smc5/6 signals in the dsDNA regions in the "Buffer only" channel, though some annotation of different regions (dsDNA and ssDNA) would have been helpful for readers. Could the mobile fluorescent spots visible in this kymograph correspond to sliding topologically-entrapped Smc5/6 complexes? It would be very helpful if the authors could clarify these points and annotate the kymographs.

Line 173: the authors write about Smc5/6 binding to dsDNA in the absence of ATP at low force, referencing Fig. 2A that presents fluorescence kymographs. It would be very helpful to have a quantification of Smc5/6 binding under these conditions, similar to Fig. 2D.

Fig S2C: were the ATP hydrolysis data for Smc5/6 obtained in the absence of DNA? It would be helpful if the authors could clarify this in the text and discuss a comparison to the reported values for basal and DNA-stimulated ATPase activity of Smc5/6.

Is it possible to extract quantitative binding parameters (e.g., k_{on}/k_{off}) from these experiments? On several occasions the authors use plots of fluorescence intensity between two time points (T1 and T2, Figs 1D, 2B, 3D, 4C, 5B, 6C, S7C). Even though the time scale can be inferred from the scale bars in kymographs, the readability of these plots would greatly benefit from adding ticks and labels with e.g. relative time values to the x-axis.

522: the authors provide the laser power in % from maximum intensity. It would be helpful to convert them to absolute values (e.g. mW)

Reviewer #2 (Remarks to the Author):

Of the three SMC protein complexes found in eukaryotic species, the cellular roles and molecular mechanisms of the Smc5/6 complex have remained the least understood. To gain insights into the function of Smc5/6, Liu, Zhao and colleagues describe the binding properties of the eight-subunit Smc5/6 complex to single- and double-stranded regions of individual DNA molecules tethered in an optical trap. They discover that the Smc5/6 complex displays a remarkable accumulation at ssDNA-dsDNA junctions either generated by overstretching the nicked DNA molecule or at a junction that recapitulates a DNA replication fork structure. In contrast to the transient association with dsDNA regions, Smc5/6 accumulation at ssDNA-dsDNA junctions is ATP-independent, displaces the ssDNA-binding protein SSB, and most likely prevents DNA re-annealing. The authors conclude that the Smc5/6 complex has distinct binding properties for different DNA substrates to stabilize the DNA replication fork or DNA damage repair intermediates.

The strength of this study lies in the real-time observation of (single) Smc5/6 complexes binding simultaneously to ssDNA and dsDNA regions. Although the structural features of Smc5/6 complexes that determine their differential binding properties remain largely unknown (see specific comments 1–3), the striking assembly of these complex at ssDNA-dsDNA junctions is a key finding that is of immediate interest to the chromosome biology field. The manuscript is clearly written and accessible to a wide audience. It is, in principle, suitable for publication in Nature Communications once the authors have addressed the concerns outlined below.

Specific comments:

1. A central aspect of the authors' model for the recruitment of Smc5/6 to ssDNA-dsDNA junctions is the binding of ssDNA to the Smc5-6 hinge domain. It is essential that the authors challenge their model by testing whether the hinge domain is indeed responsible for this property. A mutant combination that drastically reduces the ssDNA affinity of the *S. pombe* Smc5-6 hinge has been described previously (Smc5-R609E/R615E; Alt et al., Nat Comm 2017) and it should be feasible to purify the corresponding *S. cerevisiae* Smc5/6 mutant complex and test its behavior in the optical trap setup.
2. Since the dynamic association of Smc5/6 with dsDNA regions is (a) reduced for complexes that lack the Nse1-3-4 module and (b) not affected by a mutation that prevents ATP hydrolysis, the authors conclude that 'clamping' DNA between the ATP-bound Smc5-6 head domains and the Nse1-3-4 module is responsible for stable dsDNA binding. To demonstrate that this is the case and to support the authors' conclusion that "ATP binding by the Smc5-6 head regions" contributes to dsDNA binding, the authors should test a mutant complex that is deficient for ATP binding to the Smc5-6 heads (e.g. containing a mutation in the Smc5-6 P-loops or Q-loops).
3. Along these lines, I would have expected that a (ATP-bound) complex that 'clamps' DNA would be rather able to slide along dsDNA than a (ATP-free) complex that attaches to DNA via a binding surface between the Smc5 and Nse3. Evidence for DNA binding at the Smc5-Nse3 site is very limited, based solely on in-silico docking with a structural model of very limited resolution. Is it possible, based on the authors' data, to follow the DNA association and dissociation behavior of single Smc5/6 molecules on dsDNA regions to calculate on and off rates in the absence or presence of ATP or even monitor sliding along DNA? The authors' setup should be ideally suited to obtain quantitative parameters of Smc5/6 DNA binding dynamics.
4. The authors conclude that Smc5/6 dissociation from ssDNA-dsDNA junctions is considerably slower than dissociation from ssDNA or dsDNA regions when tethered DNAs were moved to a buffer-only channel (Fig. S3). Since the starting signal intensity at the junction is significantly higher (Fig. 1F), it is not surprising that signal intensities in the ssDNA or dsDNA regions drop to background levels much more rapidly. The authors should plot relative signal intensities in Fig. S3C and ideally estimate and

compare dissociation constants from a fit of the data.

5. The authors find that five-subunit complexes that lack the Nse1-3-4 module are deficient in ATP hydrolysis (Fig. S2C). It is not clear from the description of this experiment whether ATP hydrolysis rates were determined in the presence of DNA. If ATPase rates were determined in the absence of DNA, it is not clear how why the Nse1-3-4 module should be important to boost ATP hydrolysis by the Smc5-6 head domains.

Although five-subunit complexes hydrolyze ATP at only very low rates (Fig. S1C), their association with dsDNA is still increased in the presence of ATP (Fig. 3E). Although the authors claim that ATP stimulation of dsDNA binding is lower for the five-subunit complex than for the eight-subunit complex, it is in fact equal or even higher when one compares the intensity increase in relation to the starting signal (15-fold or 6.2-fold for the five-subunit complex, 2.7-fold or 7.3-fold for the eight-subunit complex in the case of DNA-LF or DNA-HF, respectively; Fig. 3E). The authors need to discuss this discrepancy in their data.

6. The authors conclude from a decrease in the fluorescence signal of labeled SSB that the Smc5/6 complex displaces SSB (Fig. 5). To rule out that the decrease in SSB fluorescence is not simply due to bleaching of (stably bound) SSB molecules, the authors need to repeat the same experiment in the absence of Smc5/6.

7. For some experiments, the number of repeats is rather low. It is essential that the authors increase the number of repeats for key experiments and report more explicitly the reproducibility of the observations to improve transparency. For example, did the authors observe an enrichment of Smc5/6 at a distance of ~ 16 kb from one end of the DNA in Fig. 4A for all molecules imaged, or just for a fraction of molecules ($n = 5$), which were then selected to calculate intensity differences?

8. The figure legends frequently reiterate the authors' conclusion of an experiment rather than describe the data that is shown in the figure. The authors need to define error bars for each graph and indicate the number of repeats for each experiment, including the supplementary figures. Details in the figure legends (e.g., how was the gel in Fig. S1A stained?) would help the reader to gain essential information directly from the figure legends.

Reviewer #3 (Remarks to the Author):

The Smc5/6 complex is an ATP-dependent protein machine that plays key roles in the maintenance of genome stability. It stabilizes DNA replication forks and helps to resolve DNA repair intermediates. How the Smc5/6 complex associates with different types of DNA remains poorly understood. In the current study, using time-resolved single-molecule fluorescence and force microscopy, Chang et al. studied the behavior of Smc5/6 on three different types of DNA, including dsDNA, ssDNA and junction DNA formed by juxtaposed ss- and dsDNA. which highly They showed that Smc5/6 binds more tightly to junction DNA, which resembles stalled or damaged replication forks. ATP binding contributes to the association of Smc5/6 to linear dsDNA, while Nse1-3-4, but not ATP binding, contributes to Smc5/6's stable assembly onto junction DNA.

Overall, their findings provide insight into how Smc5/6 interacts with different types of DNA (including branched DNA) and are broadly consistent with the *in vivo* functions of Smc5/6 in DNA repair and replication fork maintenance. This study is experimentally well designed, and the data are convincing. Publication is recommended. The following specific points need to be addressed prior to publication.

Specific points

1. The authors speculated that the hinge domain of Smc5/6 is critical for binding to DNA junctions possibly by binding to ssDNA directly. Can they design Smc5/6 hinge domain mutants that are

deficient for ssDNA binding and test their binding to different types of DNA using their single-molecule assay.

2. In the model figure (Figure 6D), the authors depicted Smc5/6 oligomerization on ssDNA, but a single Smc5/6 molecule at junction DNA. Is there experimental support for Smc5/6 oligomerization on ssDNA? Is the stoichiometry of Smc5/6 binding to junction DNA known?

3. Is the clamped state of Smc5/6 on DNA resistant to high salt, as is the case for cohesin? The authors may wish to repeat the experiments in high-salt buffers to better differentiate the two modes (clamping vs sliding) of dsDNA binding by Smc5/6.

4. Does Smc5/6 directly recognize and load on junction DNA or does Smc5/6 first load on dsDNA and then slide to junction DNA? Is the accumulation of Smc5/6 at junction DNA due to the inability of Smc5/6 to slide past these structures? Can their data differentiate between these possibilities? Please discuss.

5. In Figure S7B, Mcm10 was used to help CMG binding to the replication fork. There was no mention of Mcm10 in the text or figure legend.

We thank the reviewers for their comprehensive and careful evaluation of our work that helped us to improve our manuscript. We have addressed each point raised by the reviewers as detailed below by providing additional data and analyses and clarification in the text. Our responses below are in blue and major changes in the text are in red. We hope our responses are satisfactory.

Reviewer #1. The manuscript describes a detailed analysis of Smc5/6 binding to different types of DNA substrates (dsDNA, ssDNA and ssDNA/dsDNA junctions) at different tensions and unravels an intriguing ability of this complex to displace SSB and prevent strand reannealing. Despite extensive studies of Smc5/6 and its critical roles in a number of key cellular processes, including DNA repair and replication, the mechanisms underlying the functions of Smc5/6 remain elusive. The new single-molecule data presented here are therefore not only of outstanding quality but also extremely timely, and they substantially further our understanding of this crucial complex. I thus strongly recommend the publication of this study at Nature Communications, and, given the competitive nature of this 'hot field', as soon as possible. I do not have major points that must be addressed, but the manuscript would benefit from the authors addressing the following minor points/questions:

We appreciate the reviewer's very positive comments, the enthusiasm for our timely and substantial findings that further our understanding of Smc5/6, and the recommendation for publication as soon as possible. We addressed each of the reviewer's comments below and in the text.

1. The authors should include a more detailed discussion of the results they obtained in relation to the demonstration of loop extrusion and DNA translocation activities for Smc5/6 in a recent preprint (ref. 20). For example, it would be helpful for readers if the authors could comment on the choice of the low-force regime (around 5 pN) compared to a force regime that permits translocation and loop extrusion (<1 pN).

Our choice of a low-force regime at around 5 pN keeps the dsDNA tether taut to facilitate single-molecule imaging of Smc5/6 binding and diffusion, while still sufficiently low to maintain B-form dsDNA structure. The choice of <1 pN force used in the BioRxiv preprint was required to detect the loop extrusion activity of Smc5/6, which is not the focus of our paper. SMC loop extrusion was observed in a low-salt buffer (50 mM KCl) and the physiological relevance of Smc5/6 loop extrusion remains to be fully established. We have added a note in the revised manuscript about the choice of the low-force regime (Lines 95-97).

2. There seems to be an apparent contradiction between the circular DNA entrapment results, the statement that "ATP enhancement of Smc5/6 binding to dsDNA seen here at the single-molecule level is consistent with biochemical data showing that ATP promotes...topological entrapment of circular dsDNA" (line 174) and the statement that "... signal at linear dsDNA... rapidly dropped to background levels" (line 156). It seems like a dsDNA substrate tethered between two microspheres should not permit a ring to slide off the ends, just like circular DNA, and therefore topological binding is expected to last very long. For that reason, the fast dissociation of Smc5/6-dsDNA complex seems to argue against topological entrapment. The data in Fig. S3A seem to feature many long-lasting Smc5/6 signals in the dsDNA regions in the "Buffer only" channel, though some annotation of different regions (dsDNA and ssDNA) would have been helpful for readers. Could the mobile fluorescent spots visible in this kymograph correspond to sliding topologically entrapped Smc5/6 complexes? It would be very helpful if the authors could clarify these points and annotate the kymographs.

We apologize for the confusion. The reviewer correctly pointed out that there existed a population of long-lasting Smc5/6 complexes in the buffer-only channel shown in the original Fig. S3A, which likely corresponds to the topologically entrapped complexes. To avoid ambiguity in assigning the different DNA regions, we conducted new experiments to focus on the behavior of Smc5/6 on dsDNA in a low-force regime wherein no ssDNA or ss-dsDNA junctions were generated (described in detail in our response to Comment #5 below). Consistent with the data shown in the original Fig. S3A, we observed two populations of Smc5/6 when incubated with ATP: one displayed long-lasting binding behavior on dsDNA (reflecting topologically trapped) and the other dissociated quickly upon moving into the buffer-only channel (not topologically entrapped). We have included these new results as **new Fig. S3A** in the revised manuscript and removed the original Fig. S3.

3. Line 173: the authors write about Smc5/6 binding to dsDNA in the absence of ATP at low force, referencing Fig. 2A that presents fluorescence kymographs. It would be very helpful to have a quantification of Smc5/6 binding under these conditions, similar to Fig. 2D.

We thank the reviewer for this suggestion and have quantified and plotted Cy3-Smc5/6 signals on dsDNA at low force (5 pN) (**Fig. R1** and **new Fig. 2B**). Similar to the observations made at high force, Smc5/6 binds dsDNA at low force in the absence of ATP, and ATP enhances their interaction.

Figure R1. Quantification of Cy3-Smc5/6 fluorescence on dsDNA with or without ATP at low force (5 pN). Error bars represent standard deviations. *P* value was determined by a two-tailed unpaired *t*-test with Welch's correction (**** *P* < 0.0001). The sample size for the - ATP condition was 10 tethers and for the + ATP condition was 10 tethers.

4. Fig S2C: were the ATP hydrolysis data for Smc5/6 obtained in the absence of DNA? It would be helpful if the authors could clarify this in the text and discuss a comparison to the reported values for basal and DNA-stimulated ATPase activity of Smc5/6.

The ATP hydrolysis data in the original Fig. S2C was obtained in the absence of DNA. Per the reviewer's suggestion, we measured the ATP hydrolysis rates also in the presence of a dsDNA substrate (**Fig. R2** and **modified Fig. S2C**). Consistent with a previous report (Taschner et al., 2021), the addition of dsDNA oligoes increased the ATPase activity of Smc5/6 by about two-fold. We have clarified these results in the text.

Figure R2. ATPase activities of Smc5/6 complexes. Different forms of Smc5/6 complexes were tested with and without a 72-bp dsDNA. The wild-type Smc5/6 holo-complex (WT) with and without the S6 tag exhibited similar basal ATPase activities and both exhibited an increase in the ATPase activity in the presence of DNA. The ATPase mutant form of the Smc5/6 holo-complex (EQ) and the Smc5/6 5-mer complex exhibited reduced ATPase activities.

5. Is it possible to extract quantitative binding parameters (e.g., k_{on}/k_{off}) from these experiments?

We have performed additional experiments to extract the binding kinetics of Smc5/6 on DNA. For dsDNA binding, we incubated a dsDNA tether at low force with 5 nM of Smc5/6 in the presence of ATP for 30 s and then moved the tether to a buffer-only channel. Many Smc5/6 complexes dissociated immediately, indicating that they were non-topologically bound. Some remained bound and underwent diffusive motions in the buffer-only channel, suggesting that they were topologically bound. We were able to measure the binding lifetimes of the latter population for which discrete Smc5/6 trajectories were visible, obtaining an average lifetime of 76 s. For junction DNA binding, we pulled the DNA tether to high force and repeated the same procedure as above. In contrast to dsDNA binding, Smc5/6 stably bound at the junction sites for at least 290 s. These results are shown in **Fig. R3** below and included in the revised manuscript as a new supplemental figure (**Fig. S3**). We were not able to quantify the association kinetics because Smc5/6 signals appeared instantaneously on DNA during incubation and the signals were overwhelmed by nonspecific interactions.

6. On several occasions the authors use plots of fluorescence intensity between two time points (T1 and T2, Figs 1D, 2B, 3D, 4C, 5B, 6C, S7C). Even though the time scale can be inferred from the scale bars in kymographs, the readability of these plots would greatly benefit from adding ticks and labels with e.g. relative time values to the x-axis.

We have added ticks and relative time values to the x-axis in all our kymographs.

7. 522: the authors provide the laser power in % from maximum intensity. It would be helpful to convert them to absolute values (e.g. mW).

We have provided the absolute values for the laser power in the Methods section (Lines 514-515).

Reviewer #2. Of the three SMC protein complexes found in eukaryotic species, the cellular roles and molecular mechanisms of the Smc5/6 complex have remained the least understood. To gain insights into the function of Smc5/6, Liu, Zhao and colleagues describe the binding properties of the eight-subunit Smc5/6 complex to single- and double-stranded regions of individual DNA molecules tethered in an optical trap. They discover that the Smc5/6 complex displays a remarkable accumulation at ssDNA-dsDNA junctions either generated by overstretching the nicked DNA molecule or at a junction that recapitulates a DNA replication fork structure. In contrast to the transient association with dsDNA regions, Smc5/6 accumulation at ssDNA-dsDNA junctions is ATP-independent, displaces the ssDNA-binding protein SSB, and most likely prevents DNA re-annealing. The authors conclude that the Smc5/6 complex has distinct binding properties for different DNA substrates to stabilize the DNA replication fork or DNA damage repair intermediates. The strength of this study lies in the real-time observation of (single) Smc5/6 complexes binding simultaneously to ssDNA and dsDNA regions. Although the structural features of Smc5/6 complexes that determine their differential binding properties remain largely unknown (see specific comments 1–3), the striking assembly of these complex at ssDNA-dsDNA junctions is a key finding that is of immediate interest to the chromosome biology field. The manuscript is clearly written and accessible to a wide audience. It is, in principle, suitable for publication in Nature Communications once the authors have addressed the concerns outlined below.

We thank this reviewer for a positive evaluation, stating that our key findings are of immediate interest to the chromosome biology field and that the manuscript is well written. The main concern revolves around the uncertainty of how structural features determine Smc5/6's unique DNA binding properties, which is elaborated in the specific points below. These are important points that we thank the reviewer for bringing up, and we have addressed each by adding new data and/or improving the clarity of our writing.

1. A central aspect of the authors' model for the recruitment of Smc5/6 to ssDNA-dsDNA junctions is the binding of ssDNA to the Smc5-6 hinge domain. It is essential that the authors challenge their model by testing whether the hinge domain is indeed responsible for this property. A mutant combination that drastically reduces the ssDNA affinity of the *S. pombe* Smc5-6 hinge has been described (Smc5-R609E/R615E; Alt et al., Nat Comm 2017) and it should be feasible to purify the corresponding *S. cerevisiae* Smc5/6 mutant complex and test its behavior in the optical trap setup.

How the budding yeast Smc5/6 hinge binds to ssDNA is currently unknown. Alt et al (2018) described a structure of the DNA-free hinge fragment of the fission yeast Smc5-6; unfortunately, it is insufficient to predict the ssDNA binding sites of the budding yeast Smc5/6 for several reasons. *First*, DNA-free and DNA-bound structures can be quite different as DNA binding frequently induces conformational changes. For example, DNA-free vs. dsDNA-bound Smc5/6 structures show significant differences (Yu et al 2022; Hallett et al 2022). *Second*, due to the large evolutionary distance between the two yeasts, Smc5 and 6 sequences are quite divergent. For example, Alt et al showed that two residues affect hinge fragment-ssDNA binding *in vitro*. However, only one is conserved in the budding yeast protein, suggesting that residues for ssDNA binding in the two yeasts are likely not the same. *Third*, consistent with the above point, distribution of positively charged residues in the fission yeast Smc5-6 hinge is different from that in the budding yeast hinge based on AlphaFold structural model. Based on these considerations, we believe that the right approach to test how hinge-ssDNA binding affects Smc5/6 association with junction DNA is to determine the structure of the budding yeast hinge domain bound to ssDNA. This is a major undertaking on its own and is beyond the scope of this work.

A main conclusion of our work is that Smc5/6 can stably bind to ss-dsDNA junctions. As described in the text, the only part of Smc5/6 known to bind ssDNA is its hinge. Thus, the most logical interpretation of our finding is that Smc5/6 association with ss-dsDNA junctions entails ssDNA binding by its hinge. Given the limitations described above and to avoid confusion, we have modified our wordings in the revised manuscript to explicitly state that the above statement is an interpretation consistent with the current knowledge and explain all our data, however, its validation will require direct determination of the ssDNA binding sites of the budding yeast hinge using structural and biophysical approaches in the future (Lines 353-363).

2. Since the dynamic association of Smc5/6 with dsDNA regions is (a) reduced for complexes that lack the Nse1-3-4 module and (b) not affected by a mutation that prevents ATP hydrolysis, the authors conclude that 'clamping' DNA between the ATP-bound Smc5-6 head domains and the Nse1-3-4 module is responsible for stable dsDNA binding. To demonstrate that this is the case and to support the authors' conclusion that "ATP binding by the Smc5-6 head regions" contributes to dsDNA binding, the authors should test a mutant complex that is deficient for ATP binding to Smc5-6 heads (e.g. containing a mutation in Smc5-6 P-loops or Q-loops).

Our conclusion that ATP binding by the Smc5-6 head regions contributes to dsDNA binding was based on point (b) listed above by the reviewer, and importantly also based on that ATP addition significantly increases Smc5/6 association with dsDNA (Fig. 2B, S6E in the revised manuscript). When considering both pieces of data, a logical conclusion is that the positive effect of ATP on Smc5/6-dsDNA binding is not due to its hydrolysis, but rather its binding to the Smc5/6 head region. While an ATP binding mutant would reinforce this conclusion, we do not believe that it is required for our conclusion. Generating and biochemically testing these mutated forms of Smc5/6 as well as subjecting the mutant complex to single-molecule experiments and data analyses is a significant undertaking that we plan to pursue in a future project.

The statement that “clamping DNA between the ATP-bound Smc5-6 head domains and the Nse1-3-4 module is responsible for stable dsDNA binding” is meant to be an interpretation of our data (not a conclusion) and we have now made this point more explicit. We favor this interpretation as it is consistent with the recently published DNA-bound Smc5/6 structure (Yu et al 2022) and multiple pieces of data presented in this work, including points (a) and (b) listed by the reviewer. We have now modified the text to emphasize that this interpretation can best explain our data based on the currently available Smc5/6-dsDNA structure, and its validation requires future studies using mutants that affect the DNA clamp formation (Lines 326-335).

3. Along these lines, I would have expected that a (ATP-bound) complex that ‘clamps’ DNA would be rather able to slide along dsDNA than a (ATP-free) complex that attaches to DNA via a binding surface between the Smc5 and Nse3. Evidence for DNA binding at the Smc5-Nse3 site is very limited, based solely on in-silico docking with a structural model of very limited resolution. Is it possible, based on the authors’ data, to follow the DNA association and dissociation behavior of single Smc5/6 molecules on dsDNA regions to calculate on and off rates in the absence or presence of ATP or even monitor sliding along DNA? The authors’ setup should be ideally suited to obtain quantitative parameters of Smc5/6 DNA binding dynamics.

We thank the reviewer for this suggestion. We have performed new experiments to characterize the binding dynamics of Smc5/6 on dsDNA in the absence or presence of ATP. We first incubated 5 nM of Cy3-Smc5/6 with a dsDNA tether at low force for 30 s and then moved the tether to a buffer-only channel. In the presence of ATP, a population of Smc5/6 complexes remained bound to DNA and indeed underwent sliding along the DNA as the reviewer correctly predicted (Fig. R3A). We were able to measure the lifetimes of these sliding Smc5/6 trajectories and obtained an average lifetime of 76 s. We also observed that some Smc5/6 complexes dissociated immediately in the buffer-only channel, indicating that they were non-topologically bound. In the absence of ATP, almost all Smc5/6 binding events fell into this latter category, and the few sliding complexes lasted much shorter compared to the +ATP condition (30 s vs. 76 s) (Fig. R3B, R3D). We were not able to quantify the association kinetics because many Smc5/6 complexes appeared instantaneously on dsDNA during incubation and they displayed high diffusivity, which prevented us from resolving individual complexes. We have added the new results in the revised manuscript (new Fig. S3; Lines 175-180).

Figure R3. Smc5/6 dissociation kinetics on dsDNA and junction DNA. (A) Representative kymograph of a λ DNA tether held at 5 pN being incubated with 5 nM Cy3-Smc5/6 (green) and 2 mM ATP for 30 s (red bar) before being moved to the buffer channel (orange arrow) where the lifetimes of Smc5/6 molecules were measured. (B) The same procedure as in panel A except that ATP was not included. (C) Representative kymograph depicting a dsDNA tether held at high force being incubated with 5 nM Cy3-Smc5/6 (green) for 30 s (red bar) and then moved to the buffer channel (orange arrow). White brackets indicate photobleaching events within an Smc5/6 streak. (D) Dot plots of the lifetime of Cy3-Smc5/6 signals on (left) dsDNA_{LF} \pm ATP and (right) dsDNA versus Junction DNA in the presence of 2 mM ATP. Error bars represent standard deviations. *P* values were determined by a two-tailed unpaired *t*-test with Welch’s correction (*****P*<0.0001). The sample sizes are: dsDNA_{LF} - ATP (*n*=46); dsDNA_{LF} + ATP (*n*=116); and Junction_{HF} + ATP (*n*=109), where *n* indicates the number of Smc5/6 streaks measured.

4. The authors conclude that Smc5/6 dissociation from ssDNA-dsDNA junctions is considerably slower than dissociation from ssDNA or dsDNA regions when tethered DNAs were moved to a buffer-only channel (Fig. S3). Since the starting signal intensity at the junction is significantly higher (Fig. 1F), it is not surprising that signal intensities in the ssDNA or dsDNA regions drop to background levels much more rapidly. The authors should plot relative signal intensities in Fig. S3C and ideally estimate and compare dissociation constants from a fit of the data.

We appreciate this comment. To avoid ambiguity in assigning the DNA regions, we conducted new experiments to separately measure the dissociation kinetics of Smc5/6 from dsDNA versus junction DNA. The dissociation kinetics from dsDNA is described above (Comment #3). To determine the dissociation kinetics at junction DNA, we stretched dsDNA to high force to create junction sites, incubated the tether with 5 nM of Cy3-Smc5/6 and ATP for 30 s, and then moved the tether to a buffer-only channel. The Smc5/6 streaks at junctions (easily identified by their brightness and stationary nature) lasted for at least 290 s (Fig. R3C, R3D). Considering possible dye photobleaching, this number represents a lower bound of Smc5/6's lifetime on junction DNA. These data support our conclusion that Smc5/6 dissociation from junction DNA is much slower than from dsDNA, and they are now included in the revised manuscript as **new Fig. S3**.

5. The authors find that five-subunit complexes that lack the Nse1-3-4 module are deficient in ATP hydrolysis (Fig. S2C). It is not clear from the description of this experiment whether ATP hydrolysis rates were determined in the presence of DNA. If ATPase rates were determined in the absence of DNA, it is not clear how why the Nse1-3-4 module should be important to boost ATP hydrolysis by the Smc5-6 head domains.

The ATP hydrolysis data in the original Fig. S2C were obtained in the absence of DNA. Our interpretation of the requirement of Nse1-3-4 for intrinsic ATP hydrolysis by Smc5/6 is based on structural observations that Nse1-3-4 binds to the head regions regardless of DNA through different binding interfaces (Hallett et al 2022; Yu et al 2022). Thus, this subcomplex could in principle modulate head engagement and consequently head-mediated ATP hydrolysis.

Although five-subunit complexes hydrolyze ATP at only very low rates (Fig. S1C), their association with dsDNA is still increased in the presence of ATP (Fig. 3E). Although the authors claim that ATP stimulation of dsDNA binding is lower for the five-subunit complex than for the eight-subunit complex, it is in fact equal or even higher when one compares the intensity increase in relation to the starting signal (15-fold or 6.2-fold for the five-subunit complex, 2.7-fold or 7.3-fold for the eight-subunit complex in the case of DNA-LF or DNA-HF, respectively; Fig. 3E). The authors need to discuss this discrepancy in their data.

We apologize for this confusion. The 5-mer binding to dsDNA in the absence of ATP was very low, approaching the background level (Fig. 3C). As such, calculating the fold change can be associated with large uncertainties. We have thus removed the assertion that ATP stimulation of dsDNA binding is lower for the 5-mer than for the 8-mer complex. We have adjusted our statement to focus on the fact that the 5-mer has reduced dsDNA binding activities compared to the 8-mer in both -ATP and +ATP conditions (Lines 219-221).

Given that our data suggest that ATP binding enhances dsDNA binding by Smc5/6 (see Comment #2 above), it is possible that even though the 5-mer has a reduced ATP hydrolysis activity, its ATP binding activity is less disrupted, thus still displaying some ATP-mediated enhancement of dsDNA association.

6. The authors conclude from a decrease in the fluorescence signal of labeled SSB that the Smc5/6 complex displaces SSB (Fig. 5). To rule out that the decrease in SSB fluorescence is not simply due to bleaching of (stably bound) SSB molecules, the authors need to repeat the same experiment in the absence of Smc5/6.

We have performed the experiment suggested by the reviewer, quantifying the fluorescence signals of labelled SSB in the absence of Smc5/6. When comparing SSB streaks without Smc5/6 against those with Smc5/6 (normalized to the intensity of their first timepoints), we found that the average SSB fluorescence intensity decayed slower when Smc5/6 was absent while peak SSB intensity was delayed when Smc5/6 was present (Fig. R4). These observations are in general agreement with our interpretation that Smc5/6 can compete with SSB for ssDNA binding. However, because of the noticeable photobleaching in the absence of Smc5/6, we refrained from making the conclusion that Smc5/6 can displace SSB from ssDNA. Instead, we toned down our conclusion by stating that Smc5/6 has the ability to assemble on SSB-bound ssDNA, which is unambiguously supported by our data. We thank the reviewer for helping us solidify our conclusions.

Figure R4. Average fluorescence intensity of SSB streaks at junction DNA sites over time in the presence (red) or absence (black) of Smc5/6. Intensities are normalized to the first timepoints. The time lag between T1 and T2 is 160 s. Stars indicate peak SSB intensities. The SSB (+Smc5/6) group contained 18 streaks and the SSB (-Smc5/6) group contained 12 streaks.

7. For some experiments, the number of repeats is rather low. It is essential that the authors increase the number of repeats for key experiments and report more explicitly the reproducibility of the observations to improve transparency. For example, did the authors observe an enrichment of Smc5/6 at a distance of ~16 kb from one end of the DNA in Fig. 4A for all molecules imaged, or just for a fraction of molecules (n = 5), which were then selected to calculate intensity differences?

We appreciate this comment. These single-molecule experiments are time-consuming and by nature low-throughput. We have increased the number of repeats for some experiments where n was low (Fig. 4 and Fig. 5). For Fig. 4, we imaged a total of 22 molecules, 10 of which showed stable binding at the 16-kb position and were used for the fluorescence intensity analysis. For Fig. 5, we increased the sample size to 18 SSB streaks. These are now reported in the figure legends.

8. The figure legends frequently reiterate the authors' conclusion of an experiment rather than describe the data that is shown in the figure. The authors need to define error bars for each graph and indicate the number of repeats for each experiment, including the supplementary figures. Details in the figure legends (e.g., how was the gel in Fig. S1A stained?) would help the reader to gain essential information directly from the figure legends.

We have modified the figure legends as per the reviewer's suggestions.

Reviewer #3. The Smc5/6 complex is an ATP-dependent protein machine that plays key roles in the maintenance of genome stability. It stabilizes DNA replication forks and helps to resolve DNA repair intermediates. How the Smc5/6 complex associates with different types of DNA remains poorly understood. In the current study, using time-resolved single-molecule fluorescence and force microscopy, Chang et al. studied the behavior of Smc5/6 on three different types of DNA, including dsDNA, ssDNA and junction DNA formed by juxtaposed ss- and dsDNA. which highly They showed that Smc5/6 binds more tightly to junction DNA, which resembles stalled or damaged replication forks. ATP binding contributes to the association of Smc5/6 to linear dsDNA, while Nse1-3-4, but not ATP binding, contributes to Smc5/6's stable assembly onto junction DNA. Overall, their findings provide insight into how Smc5/6 interacts with different types of DNA (including branched DNA) and are broadly consistent with the in vivo functions of Smc5/6 in DNA repair and replication fork maintenance. This study is experimentally well designed, and the data are convincing. Publication is recommended. The following specific points need to be addressed prior to publication.

We appreciate the reviewer's positive comments that we have provided new insights into Smc5/6's interactions with different forms of DNA using well-designed experiments and convincing data, and the reviewer's recommendation for publication. Below we address each of the reviewer's comments in detail.

1. The authors speculated that the hinge domain of Smc5/6 is critical for binding to DNA junctions possibly by binding to ssDNA directly. Can they design Smc5/6 hinge domain mutants that are deficient for ssDNA binding and test their binding to different types of DNA using their single-molecule assay.

We appreciate this question. While we agree that hinge domain mutants would allow us to directly test our model, the mechanism of how budding yeast Smc5/6 hinge domains bind to ssDNA is not currently known Alt et al (2018) described a structure of the DNA-free hinge fragment of the fission yeast Smc5-6; unfortunately, it

is insufficient to predict the ssDNA binding sites of the budding yeast Smc5/6 for several reasons. *First*, DNA-free and DNA-bound structures can be quite different as DNA binding frequently induces conformational changes. For example, DNA-free vs. dsDNA-bound Smc5/6 structures show significant differences (Yu et al 2022; Hallett et al 2022). *Second*, due to the large evolutionary distance between the two yeasts, Smc5 and 6 sequences are quite divergent. For example, Alt et al showed that two residues affect hinge fragment-ssDNA binding *in vitro*. However, only one is conserved in the budding yeast protein, suggesting that residues for ssDNA binding in the two yeasts are likely not the same. *Third*, consistent with the above point, distribution of positively charged residues in the fission yeast Smc5-6 hinge is different from that in the budding yeast hinge based on AlphaFold structural model. Based on these considerations, we believe that the right approach to test how hinge-ssDNA binding affects Smc5/6 association with junction DNA is to determine the structure of the budding yeast hinge domain bound to ssDNA. This is a major undertaking on its own and is beyond the scope of this work. Given the limitation described above, we have now modified our writing to emphasize that our speculation is consistent with the current knowledge that Smc5/6 hinge region binds to ssDNA, and can explain all our data; however, its validation will require direct determination of the ssDNA binding sites within the budding yeast hinge regions using structural and biophysical approaches in the future (Lines 353-363).

2. In the model figure (Figure 6D), the authors depicted Smc5/6 oligomerization on ssDNA, but a single Smc5/6 molecule at junction DNA. Is there experimental support for Smc5/6 oligomerization on ssDNA? Is the stoichiometry of Smc5/6 binding to junction DNA known?

Thanks for raising this question. In Fig. 6D, we intended to convey the notion that there are more copies of Smc5/6 at junction regions next to a long ssDNA flap than at replication forks that only contained a short stretch of ssDNA. This is based on the comparison between the fluorescence intensities of Smc5/6 foci at those regions. To better quantify the Smc5/6 stoichiometry, we examined the Smc5/6 fluorescence signals more closely. We noticed that the Smc5/6 streaks at junction regions with a long ssDNA flap often displayed stepwise decreases in fluorescence over time (white brackets in Fig. R3C), presumably due to fluorophore photobleaching. Assuming the photon count loss for each step corresponds to one Smc5/6 complex (site-specifically labeled, one fluorophore per complex), we estimated ~3 photons/frame per Cy3-Smc5/6 (Fig. R5). In Fig. 4C, we show that the Smc5/6 intensity at replication forks is also ~3 photons/frame, suggesting that one Smc5/6 complex was residing at the replication fork junction.

With this information, we then went back to estimate the number of Smc5/6 complexes at DNA junctions with a long ssDNA flap during stretching experiments (e.g., Fig. 1D and 2C). We estimate that up to 25 complexes were residing at the junction regions by the end of their time courses. The additional Smc5/6 complexes were likely associating with the ssDNA flap as depicted in the model. We have now included this analysis in the revised manuscript (Lines 266-272).

Figure R5. Histogram of the fluorescence decrease per step in Cy3-Smc5/6 streaks at DNA junction regions.

3. Is the clamped state of Smc5/6 on DNA resistant to high salt, as is the case for cohesin? The authors may wish to repeat the experiments in high-salt buffers to better differentiate the two modes (clamping vs sliding) of dsDNA binding by Smc5/6.

We thank the reviewer for this excellent suggestion. We performed experiments in which Smc5/6 was loaded on dsDNA at low salt and low force and then moved to a channel containing a high-salt buffer (500 mM NaCl) in the presence of ATP. We observed long-range diffusion of Smc5/6 that traversed the entire length of 48.5-kbp lambda dsDNA and persisted for minutes, suggesting that Smc5/6 is topologically bound to dsDNA (clamping). High-salt resistance was drastically diminished in the absence of ATP as Smc5/6 immediately dissociated, consistent with our model that ATP binding is required for the clamped state of Smc5/6. These results are described below (Fig. R6) and included as a new supplemental figure (Fig. S4) in the revised manuscript.

Figure R6. Representative kymographs showing the differential ability of Smc5/6 complexes to remain bound on dsDNA in high salt depending on the ATP presence. (A) A kymograph shows the end of a 30 s incubation of a dsDNA tether held at 5 pN with 20 nM Cy3-Smc5/6 before the tether was moved to the high-salt channel (500 mM NaCl). Few if any Smc5/6 remained bound to the dsDNA when ATP was absent from the incubation. (B) (Top) A kymograph shows the same protocol, but 2 mM ATP was included in the incubation. (Bottom) A Smc5/6 complex displayed sustained rapid diffusion along the same dsDNA tether as shown on top in the high-salt buffer after 400 s. Orange arrows indicate when the DNA tether had completely stopped moving within the high-salt channel and the time ticks indicate the amount of time elapsed in the high-salt buffer.

4. Does Smc5/6 directly recognize and load on junction DNA or does Smc5/6 first load on dsDNA and then slide to junction DNA? Is the accumulation of Smc5/6 at junction DNA due to the inability of Smc5/6 to slide past these structures? Can their data differentiate between these possibilities? Please discuss.

While both scenarios are formally possible, based on our results that ATP is not required for junction binding but greatly promotes dsDNA binding, dsDNA loading of Smc5/6 does not appear to be a prerequisite for junction binding. In the kymographs, we also found little evidence for Smc5/6 sliding on dsDNA and then accumulating at junction sites. Under the condition where we observed the least amount of dsDNA binding—Smc5/6 5-mer (without Nse1-3-4) and without ATP—we still observed Smc5/6 signal accumulation at junction sites (Fig. 3B). Therefore, it is most likely that Smc5/6 directly recognizes and loads onto junction DNA. We have now added a discussion on these possibilities in the revised manuscript (Lines 365-367).

5. In Figure S7B, Mcm10 was used to help CMG binding to the replication fork. There was no mention of Mcm10 in the text or figure legend.

We apologize for this omission. We have now mentioned Mcm10 in the text and figure legend.

REVIEWERS' COMMENTS

Reviewer #1 (Remarks to the Author):

The authors have thoroughly addressed all the points that were raised. I am happy to recommend publication of the manuscript as is.

Reviewer #2 (Remarks to the Author):

The authors have carefully addressed the reviewer comments in their detailed rebuttal letter. In particular, they now provide new unambiguous evidence that a population of Smc5/6 complexes that remain bound to dsDNA are able to slide along the DNA double helix (comment 3). They also added a comparison of dissociation rates at dsDNA and ssDNA-dsDNA junctions (comment 4), which supports their original conclusions. I highly appreciate that the authors have performed control experiments for fluorescence bleaching and accordingly adjusted their statements regarding the ability of Smc5/6 to displace SSB (comment 6) and the differential effect of DNA on the ATPase activities of 5-mer and 8-mer complexes (comment 5). They furthermore repeated several single-molecule experiments to considerably increase the sample size, as requested (comment 7). Finally, the authors have provided additional information in the figure legends to clarify the data presented (except for Fig. S2D).

The authors argue that identifying DNA binding sites in the Smc5/6 hinge domain (comment 1) and testing a mutant complex defective in ATP binding (comment 2) is beyond the scope of the manuscript. While I agree that solving structures of a DNA-bound *S. cerevisiae* Smc5/6 hinge domain is not feasible within the context of this revision, there might be an alternative possibility to quickly test the authors' model: The authors now present an experiment that examines Smc5/6 dsDNA binding under high salt conditions (500 mM NaCl; Figure S4). Would it be possible to repeat this experiment under conditions that produce ssDNA-dsDNA junctions (i.e., high force)? If the authors' conclusion that Smc5/6 complexes were held at the junctions due to ssDNA binding by the hinge domain, which is non-topological and hence presumably salt sensitive, Smc5/6 localization to the junctions should drastically decrease upon addition of high salt buffer.

I would like to encourage the authors to add this control experiment, if technically feasible. Apart from that, I strongly recommend publication of the work in Nature Communications.

Reviewer #3 (Remarks to the Author):

The authors have addressed most of my concerns. They could not identify DNA-binding-deficient hinge mutants, but presented good reasoning about why this could not be done. Given the stiff competition they face, I would support the publication of the revised manuscript without this key piece of evidence.

Reviewer #2

The authors have carefully addressed the reviewer comments in their detailed rebuttal letter. In particular, they now provide new unambiguous evidence that a population of Smc5/6 complexes that remain bound to dsDNA are able to slide along the DNA double helix (comment 3). They also added a comparison of dissociation rates at dsDNA and ssDNA-dsDNA junctions (comment 4), which supports their original conclusions. I highly appreciate that the authors have performed control experiments for fluorescence bleaching and accordingly adjusted their statements regarding the ability of Smc5/6 to displace SSB (comment 6) and the differential effect of DNA on the ATPase activities of 5-mer and 8-mer complexes (comment 5). They furthermore repeated several single-molecule experiments to considerably increase the sample size, as requested (comment 7). Finally, the authors have provided additional information in the figure legends to clarify the data presented (except for Fig. S2D).

The authors argue that identifying DNA binding sites in the Smc5/6 hinge domain (comment 1) and testing a mutant complex defective in ATP binding (comment 2) is beyond the scope of the manuscript. While I agree that solving structures of a DNA-bound *S. cerevisiae* Smc5/6 hinge domain is not feasible within the context of this revision, there might be an alternative possibility to quickly test the authors' model: The authors now present an experiment that examines Smc5/6 dsDNA binding under high salt conditions (500 mM NaCl; Figure S4). Would it be possible to repeat this experiment under conditions that produce ssDNA-dsDNA junctions (i.e., high force)? If the authors' conclusion that Smc5/6 complexes were held at the junctions due to ssDNA binding by the hinge domain, which is non-topological and hence presumably salt sensitive, Smc5/6 localization to the junctions should drastically decrease upon addition of high salt buffer.

I would like to encourage the authors to add this control experiment, if technically feasible. Apart from that, I strongly recommend publication of the work in *Nature Communications*.

Response: We thank the reviewer for acknowledging that we have carefully addressed their comments and for recommending the publication of our work. We also appreciate the suggested additional experiment to test the salt sensitivity of ssDNA-dsDNA junction binding by Smc5/6. We would like to point out that our conclusions regarding the multiple distinct DNA binding modes by Smc5/6 are independent of the outcome of this experiment. We made it clear in our revised manuscript that elucidating the molecular mechanism of different Smc5/6-DNA binding modes requires further experiments. We wish to report results from the above suggested experiment together with other experiments (such as hinge domain mutants) in a future publication.